# Intrinsic Disorder in Tetratricopeptide Repeat Proteins

**DOI:** 10.3390/ijms21103709

**Published:** 2020-05-25

**Authors:** Nathan W. Van Bibber, Cornelia Haerle, Roy Khalife, Bin Xue, Vladimir N. Uversky

**Affiliations:** 1. Department of Molecular Medicine Morsani College of Medicine, University of South Florida, 12901 Bruce B. Downs Blvd., Tampa, FL 33612, USA; nvanbibber@mail.usf.edu (N.W.V.B.); chaerle@mail.usf.edu (C.H.); roykhalife@mail.usf.edu (R.K.); 2Department of Cell Biology, Microbiology and Molecular Biology, School of Natural Sciences and Mathematics, College of Arts and Sciences, University of South Florida, Tampa, FL 33620, USA; binxue@usf.edu; 3USF Health Byrd Alzheimer’s Research Institute, Morsani College of Medicine, University of South Florida, 12901 Bruce B. Downs Blvd., Tampa, FL 33612, USA; 4Institute for Biological Instrumentation, Russian Academy of Sciences, Federal Research Center “Pushchino Scientific Center for Biological Research of the Russian Academy of Sciences”, 4 Institutskaya St., Pushchino, 142290 Moscow Region, Russia

**Keywords:** tetratricopeptide repeat, intrinsically disordered protein, intrinsically disordered protein region, protein-protein interaction, posttranslational modification

## Abstract

Among the realm of repeat containing proteins that commonly serve as “scaffolds” promoting protein-protein interactions, there is a family of proteins containing between 2 and 20 tetratricopeptide repeats (TPRs), which are functional motifs consisting of 34 amino acids. The most distinguishing feature of TPR domains is their ability to stack continuously one upon the other, with these stacked repeats being able to affect interaction with binding partners either sequentially or in combination. It is known that many repeat-containing proteins are characterized by high levels of intrinsic disorder, and that many protein tandem repeats can be intrinsically disordered. Furthermore, it seems that TPR-containing proteins share many characteristics with hybrid proteins containing ordered domains and intrinsically disordered protein regions. However, there has not been a systematic analysis of the intrinsic disorder status of TPR proteins. To fill this gap, we analyzed 166 human TPR proteins to determine the degree to which proteins containing TPR motifs are affected by intrinsic disorder. Our analysis revealed that these proteins are characterized by different levels of intrinsic disorder and contain functional disordered regions that are utilized for protein-protein interactions and often serve as targets of various posttranslational modifications.

## 1. Introduction

The predominant “lock-and-key” model of protein functionality, according to which the biological function of a protein is determined by its highly ordered structure defined by its unique amino acid sequence [1], has obscured the fact that proteins without unique structure might also have biological functions. Intrinsic disorder (ID) in proteins, a phenomenon that describes the lack of a stable, ordered secondary and/or tertiary structure in functional proteins, had been largely ignored for decades after its first appearance in a scientific publication in 1936 [2]. This ignorance of the importance of ID resulted, in part, from the use of numerous unrelated terms associated with the same phenomenon and has been a contributing factor in keeping ID in obscurity for so long. As a result of the lack of consistent nomenclature, whenever an ID protein (IDP) or protein region (IDPR) was described, it was taken as a new discovery and looked upon as an odd exception to the classic protein structure-function paradigm, but was not identified as evidence towards the phenomenon of ID.

The emergence of informatics in general, with its ability to make information and its analysis more readily available, and bioinformatics in particular, by providing sophisticated databases and tools for protein analyses, led to a breakthrough in ID biology. Since the beginning of the 21st century, the number of publications about IDPs and IDPRs has skyrocketed, led primarily by publications from four research groups [3,4,5,6]. Over the last couple of decades IDPs and hybrid proteins containing ordered domains and IDPRs have been recognized as “one of the main types of proteins” [7], constituting a very sizable and functionally significant class of proteins [5,8,9,10,11,12,13]. Disorder content in proteomes increases as organism complexity increases [8,9,10,12,14], indicating that IDPs are heavily involved in, and often a crucial factor for, a multitude of biological processes [3,5,6,15,16,17,18,19], and the interaction between IDPRs and structured domains can increase functional versatility [20]. Furthermore, IDPs/IDPRs are characterized by a very complex and heterogeneous spatiotemporal structural organization, possessing regions that are ordered or disordered to different degree. Examples include foldons (independent foldable units of a protein), inducible foldons (disordered regions that can fold at least in part due to the interaction with binding partners), non-foldons (non-foldable protein regions), semi-foldons (regions that are always in a semi-folded form), and unfoldons (ordered regions that have to undergo an order-to-disorder transition to become functional) [21,22,23,24].

Astonishingly, given its only very recent acceptance into the scientific world, the natural abundance of disorder in the proteome has been shown to be at approximately 60%, meaning approximately 60% of proteins in eukaryotes display at least some degree of disorder [5,8,9,10,11,12,13,25,26,27,28,29,30,31,32,33,34,35,36,37,38,39,40]. The lack of a stable or ordered tertiary structure in IDPs/IDPRs enables their structural plasticity and binding promiscuity which make intrinsic disorder a major factor in protein-protein-interactions [5,11,21,23,41,42], a process that is essential for various biological functions. Furthermore, complex and highly heterogeneous structural ‘anatomy’ of IDPs/IDPRs contributes to their specific molecular ‘physiology’, where differently (dis)ordered structural elements possess well-defined specific functions [43], thereby allowing a protein molecule to be multifunctional (i.e., involved in interaction with, regulation of, and controlled by multiple structurally unrelated partners) [44].

Nevertheless, these same features that are functionally advantageous in tightly regulated disordered proteins can become detrimental, when ID in proteins runs amok. Referred to as D^2^ phenomenon [45,46], disorder in intrinsically disordered proteins is strongly associated with a number of diseases, the most devastating diseases of our time, as the majority of human cancer-associated proteins [47], as well as many proteins associated with amyloidoses [48], cardiovascular disease [49], diabetes [50], and neurodegeneration [51,52] are either completely disordered or contain long IDPRs.

An important feature of IDPs/IDPRs is the frequent presence of the tandem repeats in their amino acid sequences. Tandem repeats correspond to a specific bias in the amino acid sequence, where a pattern of one or more amino acid is repeated, and these repetitions are either directly adjacent to each other or are located in a very close proximity. Nature often uses tandem repeats in “scaffolding proteins” to promote protein-protein interactions and regulation of many key signaling pathways; i.e., biological functions central for all living cells. In fact, scaffolding proteins are modular proteins capable of interaction with multiple members of a signaling pathway, tethering them into complexes, and often modulating and regulating the function of these associated proteins. In this way, scaffold proteins are responsible for the assembly and regulation of multimolecular signaling complexes. Therefore, it is not surprising that there are several families of protein repeats including: Armadillo repeats [53], Ankyrin repeats [54], HEAT repeats [55], leucine-rich repeats [56], Kelch-like repeats [57], WD40 (also known as WD or β-transducin) repeats [58], Pentatricopeptide repeats [59], and Tetratricopeptide Repeats (TPR). It was shown that many repeat-containing proteins are characterized by high levels of intrinsic disorder, and that many protein tandem repeats can be intrinsically disordered [60,61,62,63,64,65].

TPR proteins contain between two and 20 repeats of the TPR motif consisting of 34 amino acids. Despite the lack of fully invariant residues in such domains, they are characterized by a degenerate consensus sequence containing a pattern of small and large hydrophobic amino acids [66], which can be identified by most general sequence analysis programs [67], as well as by a specialized computational tool, TPRpred [68,69]. What distinguishes the TPR domain most from other classes of repeats is its ability to stack repeats continuously one upon the other (where the helices within each repeat stack together with helices in adjacent TPRs to form a right-handed superhelix [70]), with these repeats then affecting interaction with their binding partners either sequentially or in combination, making TPRs “much better suited to be bi-functional and multi-functional molecules, where complex interactions between molecules are programmed in line” [71]. These distinct properties of TPR containing proteins make them a promising field of research particularly in medical, pharmaceutical and biotechnological applications. In addition to their inherent binding plasticity and functional diversity, the basic TPR scaffold can be redesigned to modulate binding specificity and/or affinity toward desired peptide ligands [66] and TPR containing proteins can be inhibited by designed ligands [67]. This designability is illustrated by Figure 1, where crystal structures of proteins containing two, three, and 20 copies of the consensus TPR motifs are shown.

TPR domains are present in a number of proteins, where they serve as versatile interaction modules and mediators of the formation of multiprotein complexes [67]. TPR proteins are found in all eukaryotic and prokaryotic organisms, where they provide important functions in regulating diverse biological processes (e.g., biomineralization, protein import, vesicle fusion, organelle targeting, and pathogenesis of bacteria) [67]. Although TPR proteins are traditionally associated with binding of short linear peptide motifs, in fact, they utilize a wide range of molecular recognition modes, being able to interact with short linear peptides and large globular domains [71]. As a result, they are involved in a wide spectrum of cellular processes. For example, by mediating protein-protein interactions and forming multiprotein complexes with cellular and viral proteins through their multiple TPR motifs, interferon-induced protein with tetratricopeptide repeats (IFIT) play a number of important roles in host innate immunity, antiviral immune response, virus-induced translation initiation, replication, double-stranded RNA signaling, and recognition of pathogen-associated molecular patterns (PAMPs) [74,75]. Although typically TPR proteins do not have enzymatic activities, ubiquitously transcribed X-chromosome tetratricopeptide repeat protein (UTX) is known to serve as H3K27 demethylase catalyzing the demethylation of di- and tri-methylated histone H3 lysine 27 (H3K27) [76], thereby being linked to homeotic gene expression, embryonic development, and cellular reprogramming [77], whereas a co-chaperone CHIP (the C terminal Hsp70 binding protein), in addition to using its TPR domains to interact with Hsp70 and Hsp90 chaperones, serves as an E3 ubiquitin ligase utilizing a modified RING finger domain (U-box) [78]. Some TPR proteins might act as chaperones (e.g., the members of the UCS (UNC-45/CRO1/She4p) family of proteins are specific chaperones for the folding, assembly, and function of myosin [79]). Among various biological functions ascribed to the TPR proteins are: control of proteins responsible for organization and homeostasis of the endoplasmic reticulum [80]; regulation of the biosynthesis of the photosynthetic apparatus via involvement in almost all of the steps crucial for biogenesis of the thylakoid membranes [81]; assistance in the *de novo* assembly and stability of a multi-component pigment-protein complex, photosystem II (PSII) [82]; the quality control of secretory and membrane proteins mislocalized to the cytosol [83]; control of limb regeneration in crickets [84] and plastid protein import [85]. Also, in biotechnology, because of their binding versatility and designability, TPR repeats are used for the creation of repeat protein scaffolds serving as a basis of the biomolecular templating of functional hybrid nanostructures [86,87].

The particular structure of the TPR motif, a helix-turn-helix conformation, linked to the next repeat by a short loop [71] forming an overall super-helix structure [67], has been characterized as an “exceptionally versatile fold capable of mediating protein-protein interactions via varied mechanisms” [71]. For example, a short peptide may bind to the TPR concave groove and an extended peptide may bind to the TPR concave groove. The TPR motif is capable of binding interacting partners of various secondary structures, and TPR proteins are known to be co-chaperones, binding to chaperones and assisting them in their task of helping proteins to fold.

TPR-containing proteins share many characteristics with hybrid proteins containing ordered domains and IDPRs. Like other IDPR-containing proteins TPR-containing proteins are naturally abundant (using bioinformatics tools, more than 5000 TPR-containing proteins were found in different organisms [67]) and are involved in many biological functions, including protein-protein interactions. Furthermore, similar to other IDPs/IDPRs, mutations in TPR-containing proteins have been associated with a variety of human diseases, such as chronic granulomatous disease or Leber’s congenital amaurosis [66,67]. Therefore, in this paper we analyzed the degree to which proteins containing TPR motifs are affected by intrinsic disorder. Data on the prevalence of intrinsic disorder in human TPR proteins may suggest yet another possible route for the sophisticated pathogenicity attributed to these proteins.

## 2. Results and Discussion

### 2.1. Per-Residue Intrinsic Disorder Predisposition of Human TPR Proteins

In this study, we analyzed 166 human proteins containing TPR domains. The amino acid sequence of these proteins ranged in length from 151 to 4471 residues. Figure 2A presents the length distribution of 166 human TPR proteins and shows that most abundant are proteins are between 500–700 residues in length.

Figure 2B shows that human TPR proteins contain from one to 28 TPR repeats, with majority proteins possessing three repeats. Peculiarities of the distribution of TPR repeats within the amino acid sequences of TPR proteins are illustrated by Figure 2C, which indicates although TPR repeats can be found in different parts of proteins, they are less likely to be C-terminal.

More than half of human TPR proteins (85) contain only TPR repeats and do not include other functional domains. However, many TPR proteins, in addition to TPR repeats, include other protein interaction modules, such as coiled-coil regions (11 proteins), ankyrin repeats (seven proteins), WD repeats (four proteins), HEAT repeats (two proteins), armadillo repeats (one protein), SH3 domains (four proteins), zinc finger motifs of different type (eight proteins), as well as the bipartite CS domains, which are ~100-residue protein-protein interaction modules named after CHORD-containing proteins and SGT1 (two proteins), a heat shock chaperonin-binding motif STI1 (three proteins), RanBD domains that interact with and stabilize the GTP-bound form of Ran (the Ras-like nuclear small GTPase) (seven proteins), and GoLoco or G-protein regulatory (GPR) motif found in various G-protein regulators (two proteins). There are also some catalytic domains in several of these proteins, such as PPIase FKBP-type domain (six proteins), SAM-dependent MTase PRMT-type 1 domain, helicase domain, serine/threonine-protein phosphatase domain, PI3K/PI4K kinase domain, and the Fe2OG dioxygenase domain, suggesting that some enzyme utilize TPR motifs for interaction with their partners.Disorder predisposition of human TPR proteins was evaluated using a set of disorder predictors from the PONDR family. The use multiple computational tools for prediction of intrinsic disorder in proteins is an accepted practice in the field. Since different computational tools use different attributes (such as amino acid composition, hydropathy, sequence complexity, etc.) and models for to calculate a disorder predisposition score for every amino acid residue in a query protein, it is a common situation, when different tools will generate rather different outputs.

There is no an accepted consensus of which disorder predictor is the best in evaluating disorder predisposition of a query protein. In reality, since different computational tools are sensitive to different disorder-related aspects of the amino acid sequence, all of them contain some useful information.

The per-residue disorder predisposition scores are on a scale from 0 to 1, where values of 0 indicate fully ordered residues, and values of 1 indicate fully disordered residues. Values above the threshold of 0.5 are considered disordered residues, whereas residues with disorder scores between 0.25 and 0.5 are considered flexible residues. For each protein, per-residue disorder predisposition values are summarized to produce global disorder predictions for the protein. The predicted average disorder score is defined by taking the arithmetic mean over all the residues to give the overall mean disorder score on the same 0 to 1 scale. The predicted percent disorder for each protein is defined as the ratio of the number of disordered residues (i.e., residues with the predicted disorder scores greater than 0.5) to the total number of residues and is given as a percentage. Based on their levels of average intrinsic disorder score, proteins were classified as highly ordered (average disorder score < 0.25), moderately disordered (average disorder score between 0.25 and 0.5) and highly disordered (average disorder score ≥ 0.5). Similarly, proteins can be classified based on their percent of predicted disordered residues (PPDR). Here, two arbitrary cutoffs for the levels of intrinsic disorder are used to classify proteins as highly ordered (PPDR < 10%), moderately disordered (10% ≤ PPDR < 30%) and highly disordered (PPDR ≥ 30%) [88]. The frequency distribution for the PPDR score for all three predictors are shown in Figure 3 and corresponding data are outlined below.

Our analysis revealed that for disorder predictions generated by PONDR^®^ VLXT for 166 human TPR proteins, the average predicted disorder score ranged from 0.11 to 0.67 with a mean of 0.34. The overall percent of disorder ranged from 3.1% to 73.2% with a mean of 30.2% (Figure 3A). The longest disordered region in query proteins ranged from 10 residues to 348 residues with a mean of 61 residues (Table 1).

According to the PONDR^®^ VSL2-based analysis, the average prediction score for 166 human TPR proteins ranged from 0.25 to 0.79 with a mean of 0.45. The overall percent disorder score ranged from 11.9% to 90.4% with a mean of 37.9% (Figure 3B). The longest disordered region ranged from 11 residues to 818 residues with a mean of 129 residues (Table 1). PONDR^®^ FIT is a meta-predictor that incorporates predictions from several different sources [89]. For PONDR^®^ FIT, the average prediction score ranged from 0.16 to 0.68 with a mean of 0.31. The overall percent disorder score ranged from 1.9% to 74.9% with a mean of 22.3% (Figure 3C).

Curiously, Figure 2D shows that there is no obvious correlation between the number of TPR motifs in human TPR proteins and their extent of disorder.

As it was already emphasized, based on their content of disordered residues, proteins are typically grouped into three categories: highly ordered (0%–10%), moderately disordered (10%–30%), and highly disordered (greater than 30%). A breakdown of these categories based on the outputs of all three disorder predictors utilized in this study can be seen in Figure 4. This analysis clearly shows that the majority of human TPR proteins are either moderately or highly disordered.

Since the average disorder score (ADS) of a given protein is not directly related to its PPDR value (e.g., theoretically, a protein with the PPDR of 100% might have the ADS ranging from 0.5 to 1.0; whereas a protein with the PPDR of 0% might have any ADS < 0.5), Figure 5 represents the ADS vs. PPDR plots generated for human TPR proteins based on the results of their analysis by PONDR^®^ VLXT, PONDR^®^ VSL2, and PONDR^®^ FIT. This figure provides further support to the idea that the majority of TPR proteins are either moderately or highly disordered. This analysis allowed us to select the most disordered proteins in the set, which were defined here as proteins possessing ADS ≥ 0.5 and/or PPDR ≥ 30% by any of the PONDR^®^ VLXT, PONDR^®^ VSL2, and PONDR^®^ FIT predictors or any their combination. We found that 59 (35.5%) of TPR proteins satisfy these criteria.

Agreement between the outputs of different intrinsic disorder predictors for a set of query proteins can be assessed in the form of a 3D scatter plot, where the average disorder scores (or average PPDR values) generated by disorder predictors are plotted against each other for all the proteins (Figure 6).

Since some of the human TPR proteins are associated with various diseases, to see if such disease-associated TPR proteins have stronger tendency for intrinsic disorder than proteins whose disease associations are unknown, we included information from UniProt on query protein associated disease or mutagenesis (Figure 6). These analyses clearly demonstrated that there is a rather good agreement between the outputs of PONDR^®^ VLXT, PONDR^®^ VSL2, and PONDR^®^ FIT for 166 human TPR proteins. This conclusion follows from the fact that the majority of points corresponding to human TPR proteins are located either on or in the close proximity to the diagonal in the corresponding 3D plots. Furthermore, proteins associated with disease or mutagenesis did not demonstrate exceptionally high levels of intrinsic disorder.

### 2.2. Binary Classification of the Intrinsic Disorder Status of Human TPR Proteins

Important information pertaining to the global classification of disorder status of query proteins can be obtained from the analysis of binary disorder predictors (i.e., computational tools that classify proteins as wholly ordered or wholly disordered). The usefulness of this approach is based on the principle differences in the criteria used by different binary predictors, such as the charge-hydropathy (CH) plot [4,14] and the cumulative distribution function (CDF) plot [14,90]. CH-plot utilizes information on absolute mean net charge of a query protein and its mean hydropathy as the only two parameters for classifying query proteins as proteins with substantial amounts of extended disorder (native coils and native pre-molten globules) or proteins with compact globular conformations (native molten globules and ordered proteins) [14,91]. On the other hand, CDF analysis uses the PONDR outputs to discriminate all types of disorder (native coils, native molten globules and native pre-molten globules) from ordered proteins [14]. Therefore, the combined CH-CDF plot gives an opportunity for unique assessment of intrinsic disorder in several categories, allowing predictive classification of proteins into structurally different classes [90,92,93].

To generate a corresponding CH-CDF plot, the coordinates of a query protein are calculated as a distance from the boundary in the CH-plot (Y-coordinate) and an average distance of the respective CDF curve from the CDF boundary (X-coordinate). In the CH-CDF plot, the x-axis separates proteins predicted by CH-plot to be intrinsically disordered (positive y values) or compact (negative y values), whereas the y-axis separates proteins predicted by CDF to be ordered (positive x values) or intrinsically disordered (negative x values). Therefore, proteins can be classified by their position within each quadrant of the resultant CH-CDF plot. Here, the lower-right quadrant (Q1) includes ordered proteins (i.e., those predicted as ordered and compact by both CDF and CH); the lower-left quadrant (Q2) contains proteins predicted to be disordered by CDF but compact by CH-plot (i.e., native molten globules or hybrid proteins containing sizable levels of order and disorder); the upper-left quadrant (Q3) contains proteins predicted to be disordered by both methods (i.e., proteins with extended disorder, such as native coils and native pre-molten globules); and the upper-right quadrant (Q4) contains proteins predicted to be disordered by CH-plot but ordered by CDF [92]. Figure 7 presents the results of this analysis for 166 human TPR proteins and shows that 12 of them are highly disordered, 52 have a molten globular or hybrid structure, and 100 are mostly ordered. In other words, based on these analyses, ~38.6% of human TPR proteins are expected to be globally disordered, which is close to the number generated by per-residue predictors (35.5%, see above).

### 2.3. Analysis of the Intrinsic Disorder-Based Functionality of Human TPR Proteins

It is known that there is an intricate correlation between protein intrinsic disorder and alternative splicing, where alternatively spliced exons in mRNA are overall more likely to encode IDPRs rather than ordered protein segments [31,94,95,96]. This tendency is even more pronounced in alternative exons, whose inclusion or exclusion is regulated in a tissue-specific manner [97]. Therefore, we analyzed the correlation between the number of alternative isoforms reported for human TPR proteins and their disorder status. Results of this analysis are summarized in Figure 8, which shows that although vast majority of human TPR proteins can exist in more than one alternative isoform, there is no obvious correlation between their disorder level and the number of isoforms.

An important feature of IDPs and IDPRs is their remarkable binding promiscuity [5,11,21,23,41,42]. In fact, these binding “professionals” are always complexed, invariably interacting with various partners via multiple binding modes [5,11,21,41], and forming static, semi-static, dynamic, or fuzzy complexes [23,42]. They are also capable of semi-static and dynamic polyvalent interactions [98], where multiple binding sites of one protein are simultaneously bound to multiple receptors on another protein [99]. Many IDPs/IDPRs bind partners with both high specificity and low affinity [100], and can fold at binding to their partners [15,16,101]. The degree of such binding-induced folding can be different in various systems, thereby forming complexes with broad structural and functional heterogeneity [23,42]. Furthermore, some IDPs/IDPRs serve as morphing shape-changers that are able to adopt alternate folds as a result of binding to different partners [101,102,103,104,105,106]. The binding region of such a morphing IDP can assume completely different structures in the rigidified assemblies formed by binding to divergent partners [18,41,107,108,109]. Many other IDPs/IDPRs are known to form fuzzy complexes, where significant disorder is retained, at least outside the binding interface [110,111,112,113,114,115,116,117]. Therefore, it is not surprising that many IDPs/IDPRs serve as hub proteins – nodes in complex protein-protein interaction (PPI) networks that have a very large number of connections to other nodes [106,118,119,120,121,122,123].

It seems that the aforementioned concept of “morphing shape-changers” is not directly applicable to the TPR repeats, which are structured peptide-binding motifs that define the multivalency of corresponding TPR proteins. However, one should keep in mind that in addition to these ordered repeats, many TPR proteins contain other domains, and, most importantly, functional IDPRs that can be involved in protein-protein interactions, and which, therefore, serve as the illustrative examples of the disorder-based “morphing shape-changers.” Obviously, while considering TPR proteins, it would be clearly a mistake to focus exclusively on ordered TPR repeats. Instead, these proteins should considered in their entirety.

Based on all these considerations we decided to check interactability of human TPR proteins using Search Tool for the Retrieval of Interacting Genes; STRING, http://string-db.org/. Figure 9 represents the inter-TPR PPI network generated by STRING for 161 human TPR proteins (no STRING information was available for 5 TPR proteins, UniProt IDs: O14607, Q86TZ1, Q6P2S7, Q8IZP2, and Q8NFI4) and shows that most of these proteins are involved in the formation of a rather dense PPI network. In fact, only two of the remaining 161 TPR proteins (UniProt IDs: Q5W5X9 and Q7Z3J3) are not included in this network. As a result, this network contains 159 nodes (TPR proteins) connected by 1,392 edges (PPIs). In this network, the average node degree is 17.3, and its average local clustering coefficient (which defines how close its neighbors are to being a complete clique; the local clustering coefficient is equal to 1 if every neighbor connected to a given node *N_i_* is also connected to every other node within the neighborhood, and it is equal to 0 if no node that is connected to a given node *N_i_* connects to any other node that is connected to *N_i_*) is 0.499. Since the expected number of interactions among proteins in a similar size set of proteins randomly selected from human proteome is equal to 708, this TPR internal PPI network has significantly more interactions than expected, being characterized by a PPI enrichment *p*-value of < 10^−16^. Therefore, these data indicate that the majority of human TPR proteins can interact with each other (although the confidence of this inter-family PPI network is low). Increasing STRING confidence level to 0.4 (moderate confidence, which is a default minimum required interaction score) results in reduction of the reliably interacting proteins from 159 to 124. In the resulting PPI network (see Appendix A), 124 nodes are linked by 301 edges, giving the average node degree of 3.74, average local clustering coefficient of 0.422, expected number of edges of 89 and PPI enrichment *p*-value of < 10^−16^. Further increase in confidence level results in the breakage of the network into a few small clusters and a multitude of non-interacting proteins (see Appendix A for a network with high confidence of 0.7).

Next, we looked at the global interactivity of the whole set of human TPR proteins by including a first shell interactors (i.e., proteins interacting with TPR proteins). The resulting PPI network generated by STRING is shown in Figure 10. Note that the number of interactors in STRING is limited to 500. In this analysis, the highest confidence level of 0.9 was used. The resulting interactome includes 661 nodes connected by 7655 edges. Therefore, this interactome is characterized by an average node degree of 23.2 and it shows an average local clustering coefficient of 0.67. The expected number of interactions for the set of proteins of its size is 3838, indicating this human TRP-centered PPI network has significantly more interactions than expected (PPI enrichment *p*-value is < 10^−16^). Decreasing STRING confidence level to 0.15 (this low confidence level ensures inclusion of the maximal number of TRP proteins into the inter-TPR PPI network) generated a dense network of 661 proteins connected by 35,582 interactions with the average node degree of 108, average local clustering coefficient of 0.464, expected number of edges of 23,094, and PPI enrichment *p*-value of < 10^−16^ (see Appendix A).

We also looked at the correlation between the interactability of individual TRP proteins evaluated by STRING at different confidence levels and their level of intrinsic disorder. Results of this analysis are shown in Figure 11 illustrating that no relationship was observed between the PONDR-FIT percent disorder score and the number of protein interactions.

### 2.4. Structural Properties of Human TPR Proteins

To shed some light on the structural repertoire (or structural mosaic) of human TPR proteins, we collected available information on structures of different members of this family from PDB. Results of this analysis are summarized in Figure 12, which shows structures of 56 TPR proteins and illustrates that the vast majority of human TPR proteins with known structures are mostly α-helical.

However, although it was known a priori that TPR repeats are helical, Figure 12 indicates than not all human TPR proteins are α-helical. This is because of the fact that many structures shown in Figure 12 are not representing the entire TPR proteins, but rather their crystalizable domains or regions amenable to NMR- or cryo-electron microscopy-based structural characterization. 

This fact is further illustrated by Figure 13, showing the fractions of these proteins represented by the experimentally validated ordered domains. This analysis reveals that there is a week anti-correlation between the structural coverage of 56 human TPR proteins evaluated for each protein as percent of its sequence that is mapped to all the known structures and intrinsic disorder predisposition calculated as corresponding PONDR^®^ FIT-based PPDR values. Figure 13 shows that no structure is currently available for any TPR protein with disorder content exceeding 55%. Furthermore, one can see there an expected trend, where more disordered proteins show less structural coverage. Furthermore, some of the structures shown in Figure 12 are not the TPR motifs of the corresponding proteins but rather represent other functional domains. For example, structures with the PDB IDs 1RL1, 5U9J, and 3B7X that contain noticeable levels of β-structure correspond to the bipartite CS domain (which is a ~100-residue protein-protein interaction module named after CHORD-containing proteins and SGT1) of human protein SGT1 homolog (PDB ID 1RL1), PPIase FKBP-type domain of aryl-hydrocarbon-interacting protein-like 1 (AIPL1; PDB ID: 5U9J), and another PPIase FKBP-type domain of the inactive peptidyl-prolyl cis-trans isomerase FKBP6 (PDB ID: 3B7X). Furthermore, some of the structures shown in Figure 12 contain noticeable disordered regions.

### 2.5. Functional Analysis of Most Disordered Human TPR Proteins

Finally, we conducted functional disorder analysis for 10 most disordered TPR proteins. Results of this analysis are outlined below. 

Figure 14 presents the corresponding data for the most disordered TPR protein, human tetratricopeptide repeat protein 1 (TTC1), which is characterized by the percent of predicted disordered residues (PPDR) of 73.3 ± 15.4% and the average disorder score (ADS) of 0.65 ± 0.11. The N-terminal half of this 292-residue long protein, which is involved in unfolded protein binding, is highly disordered (Figure 14A). TTC1 contains three TPRs, residues 116–149, 155–188, and 189–222 with the ADS values of 0.61 ± 0.18, 0.55 ± 0.21, and 0.49 ± 0.17, respectively, and has multiple sites of different posttranslational modifications and 7 molecular recognition features (MoRFs), disorder-based protein-protein interaction sites where disordered regions can fold at interaction with binding partners (Figure 14B). Finally, Figure 14C presents the TTC1-centered PPI network generated by STRING with the medium confidence level of 0.4. There are 72 nodes in this network, which are connected by 280 edges. This network is characterized by the average node degree of 7.78, average local clustering coefficient of 0.785, expected number of edges of 115, and PPI enrichment *p*-value of <10^−16^.

Figure 15 presents the results of functional disorder analysis of human tetratricopeptide repeat protein 36 (TTC36; PPDR = 66.1 ± 13.6%; ADS = 0.593 ± 0.063). This 189-residue-long protein related to cilium assembly has disordered N- and C-tails, and there are two long IDPRs (see Figure 15A). TTC36 contains three TPRs, residues 51-84, 86-118, and 123-156, with the ADS values of 0.51 ± 0.18, 0.56 ± 0.13, and 0.72 ± 0.10, and is predicted to have two MoRFs, residues 7-28 and 73-82 (see Figure 15B). The mostly disordered N-terminal region is missing in isoform 2, where the residues 1-59 were removed as a result of alternative splicing. Although the first MoRF is located outside the TPR-containing region, the second MoRF is positioned at the C-terminal half of this TPR (Figure 15B). Therefore, alternative splicing is removing one of the MoRFs, potentially affecting disorder-based interactability of this protein. TTC36-centered PPI network generated by STRING with the medium confidence level of 0.4 includes 27 proteins connected by 127 edges (see Figure 15C). This network is characterized by the average node degree of 9.41, average local clustering coefficient of 0.889, expected number of edges of 28, and PPI enrichment *p*-value of < 10^−16^.

Figure 16 reflects functional disorder analysis of human nuclear autoantigenic sperm protein (NASP; PPDR = 76.1 ± 3.6%; ADS = 0.711 ± 0.071). NASP is needed for cell proliferation, DNA replication, and normal cell cycle progression. This 788-residue-long protein interacts with HSP90 and H1 linker histones (via the histone-binding regions at residues 116-127, 211-244, and 469-512 [125]) and stimulates HSP90 ATPase activity. NASP has three TPRs, residues 43-76, 542-575, and 584-617 with the ADS values of 0.21 ± 0.08, 0.37 ± 0.10, and 0.28 ± 0.18, and four coiled-coil regions (residues 136–164, 460–487, 597–665, and 753–778). There are three alternatively spliced isoform of NASP. In the isoform 2, region 138–476 is missing; isoform 3 has MAMESTATAAVAAELVSADKIEDVPAPSTSADKVES → MFLLLLHLQIKWRATINLLSVTED GLHFVEYYLNRIIH substitution at 1–36 region, and isoform 4 misses the 36–99 region. Figure 16A,B show that more than 400-residue-long region (residues 100–516) located in the middle of NASP is predicted to be highly disordered. This long IDPR includes all histone-binding regions and two coiled-coil regions and is predicted to contain 11 MoRFs. Appendix A shows how alternative splicing affects disorder predisposition of human NASP. It is clearly seen that due to the removal of a long central IDPR, isoform 2 is essentially more ordered than the canonical form. Similarly, alternative splicing induced changes in disorder of the N-tail, making isoform 3 more ordered than the canonical form, and isoform 4 more disordered than the canonical NASP. Figure 16B shows that the entire NASP is heavily decorated with numerous phosphorylation, acetylation and ubiquitylation sites and contains 18 MoRFs, which incorporate more than half of the residues of this protein. Therefore, it is not surprising that NASP serves as a hub of very dense PPI network (Figure 16C) generated by STRING with the medium confidence level of 0.4. This network includes 207 proteins linked by 6969 interactions and is characterized by the average node degree of 67.3, average local clustering coefficient of 0.738, expected number of edges of 1161, and PPI enrichment *p*-value of < 10^−16^.

Figure 17 shows functional disorder profiles and interaction network of human tetratricopeptide repeat protein 31 (TTC31; PPDR = 62.2 ± 11.2%; ADS = 0.591 ± 0.097). This 519-residue-long protein contains a coiled-coil region (residues 147–197) and three TPRs, residues 305–338, 339–372, and 373–406 with the ADS values of 0.17 ± 0.12, 0.15 ± 0.06, and 0.30 ± 0.11. TTC31 has an alternatively spliced isoform, where the 281-285 region underwent REERP → ATSPC, and where the entire C-terminal half (residues 286-519) is missing. Disorder profile of TTC31 (see Figure 17A,B) is characterized by the presence of two long IDPRs (residues 82–310 and 399–519) that cover coiled-coil region and C-terminal region of last TPR. Because of the removal of the mostly ordered C-terminal region containing all TPRs by alternative splicing, the resulting isoform is highly disordered (see Appendix A). Several PTM sites and 12 MoRFs are spread over this protein (Figure 17B). TTC31-centered PPI network generated by STRING includes 69 proteins connected by 689 interactions (see Figure 17C). This network is characterized by the average node degree of 20, average local clustering coefficient of 0.813, expected number of edges of 115, and PPI enrichment *p*-value of < 10^−16^.

Figure 18 presents the results of functional disorder analysis of human Hsc70-interacting protein (HIP, or protein FAM10A1; PPDR = 62.4 ± 10.5%; ADS = 0.616 ± 0.072). This 369-residue-long protein forms homotetramers and is engaged in interaction with heat shock proteins, where one HIP homotetramer binds the ATPase domains of at least two HSC70 molecules.

HIP has three TPRs, residues 114–147, 148–181, and 182–215 with the ADS values of 0.35 ± 0.23, 0.28 ± 0.14, and 0.26 ± 0.12, and a heat shock chaperonin-binding motif, STI1 domain (residues 319–358). According to Figure 18A,B, long regions at both N- and C-termini of HIP are predicted to be disordered (residues 1–122 and 213–369), suggesting that STI1 domain is a disorder-based binding region. In agreement with this hypothesis, Figure 18B shows that HIP contains multiple PTMs sites and 9 MoRFs, one of which (residues 306–337) overlaps with STI1 domain. In the HIP-centered PPI network (Figure 18C), there are 78 nodes linked by 783 interactions. This STRING-generated network is characterized by the average node degree of 20.1, average local clustering coefficient of 0.793, expected number of edges of 180, and PPI enrichment *p*-value of < 10^−16^.

Results of intrinsic disorder analysis in 770-residue-long human tetratricopeptide repeat protein 14 (TTC14; PPDR = 56.1 ± 15.0%; ADS = 0.545 ± 0.114) are shown in Figure 19. This protein can interact with DNA and contains a conserved S1 motif with unknown function (residues 126–208), four TPRs (residues 210–243, 307–340, 342–374, and 382–415 with the ADS values of 0.59 ± 0.08, 0.21 ± 0.14, 0.29 ± 0.09, and 0.39 ± 0.18), as well as five regions with compositional bias, such as Poly-Glu (residues 393–396), Ser-rich (residues 475–525), and three Poly-Ser regions (residues 475–483, 487–497, and 580–583). In addition to the canonical form, TTC14 has two alternatively spliced isoforms. In isoform 2, the 592–653 region is changed to VYSYLFKKLTIKQPQAGPSGDIPEEGIVIIDDSSIHVTDPEDLQVGQ DMEVEDSGIDDPDHG and residues 654–770 are missing. In isoform 3, the 340-residue-long C-terminal region 431–770 is shortened to the VIPYFLLEI peptide. Figure 19A indicates that TTC14 has a highly disordered C-terminal region (residues 389–770) that incorporates all regions with compositional bias.

Furthermore, all alternative splicing-induced changes take place within this long C-terminal IDPR. Figure 19B shows that this protein has multiple PTMs and MoRFs, which are mostly concentrated within the disordered C-terminal region. According to STRING analysis (see Figure 19C), TTC14 is engaged in interaction with 40 partners. In the resulting PPI network, there are 133 edges, and it is characterized by the average node degree of 6.65, average local clustering coefficient of 0.762, expected number of edges of 44, and PPI enrichment *p*-value of < 10^−16^.

Figure 20 shows functional disorder profiles and interaction network of human tetratricopeptide repeat protein 9B (TTC9B; PPDR = 60.4 ± 15.0%; ADS = 0.590 ± 0.053). This 239-residue-long protein has a Pro-rich segment (residues 17–50) and contains two TPRs, residues 65–99 and 171–204 (ADS values of 0.63 ± 0.14 and 0.21 ± 0.16). TTV9B has an alternative splicing-generated isoform, in which the C-terminal region 143–239 is changed to a shorter sequence GTPSGGGGMGHEGRGQSGELGD LGAR GPGAEGSRAVGFSGSLQSLIKERD. Disorder is concentrated within the N-terminal half of this protein (see Figure 20A,B), which can be approximated as one long IDPR (residues 1–140). Alternative splicing-induced changes in TTC9B sequence convert this protein to a completely disordered form characterized by PPDR = 93.0 ± 6.0% and ADS = 0.787 ± 0.010 (see Appendix A). The first TPR of TTC9B overlaps with longest of three MoRFs found in this IDPR (residues 1-16, 52-98, and 140-146) and contains majority of phosphorylation sites (Figure 20B). Based on STRING analysis (Figure 20C), TTC9B interacts with 40 partners, and is located in the middle of a PPI network containing 149 edges and characterized by the average node degree of 7.27, average local clustering coefficient of 0.824, expected number of edges of 42, and PPI enrichment *p*-value of < 10^−16^.

Figure 21 shows functional disorder profiles and interaction network of human protein SMG7 (PPDR = 56.6 ± 11.7%; ADS = 0.543 ± 0.096). SMG7 is a 1137-residue-long protein that plays a role in the nonsense-mediated mRNA decay (NMD), a process by which aberrant mRNAs containing nonsense mutations and premature termination codons (PTCs) are degraded [126,127]. SMG7 recruits regulator of nonsense transcripts 1 (a.k.a. up-frame-shift suppressor 1 homolog, UPF1) to cytoplasmic mRNA decay bodies, where they interact with each other and SMG5 via their N-terminal domains to from a complex [127]. SMG7 contains two TRPs (residues 152–185 and 187–219 with the ADS values of 0.39 ± 0.26 and 0.24 ± 0.21), and Gln/Pro-rich and Ser-rich regions (residues 648–843 and 922–1015, respectively). In addition to the canonical form, SMG7 has several isoforms generated by alternative splicing. Residues 569-614 are missing in both isoform 2 and isoform 4. Also, in isoform 4, residue Glu914 is changed to the residue sequence EDPKSSPLLPPDLLKSLAALEEEEELIFSNPPD LYPALLGPLASLPGRSLF, and the 38-residue-long C-terminal region (residues 1101–1137) is changed to a longer tail KQQHGVQQLGPKRQSEEEGSSSICVAHRGPRPLPSCSLPASTFRVKFKA ARTCAHQAQKKTRRRPFWKRRKKGK. Isoform 5 is missing 42 N-terminal residues and has an E → EDPKSSPLLPPD LLKSLAALEEEEELIFSNPPDLYPALLGPLASLPGRSLF substitution at position 914.

Although X-ray crystal structure is available for the N-terminal domain of SMG7 (residues 1–497; PDB ID: 1YA0 [126]), Figure 21A,B show that C-terminal half of this protein is highly disordered. All sequence changes introduced by the alternative splicing take place within this long IDPR. Appendix A shows how alternative splicing affects disorder predisposition of human SNG7. Figure 21B indicates that the highly disordered C-terminal IDPR of this protein contains multiple sites of different PTMs (phosphorylation, methylation, acetylation, and ubiquitylation) and includes 16 MoRFs, suggesting that SNG7 can exhibit high binding promiscuity. This hypothesis is supported by Figure 21C representing STRING-generated PPI network centered at SNG7. This protein can interact with 120 partners, and its network includes 6379 edges and characterized by an average node degree of 106, average local clustering coefficient of 0.988, expected number of edges of 835, and PPI enrichment *p*-value of < 10^−16^.

Results of functional disorder analysis of human FK506-binding protein-like (FKBPL; PPDR = 53.9 ± 14.6%; ADS = 0.525 ± 0.088) are summarized in Figure 22. FKBPL, also known as WAF-1/CIP1 stabilizing protein 39 (WISp39) is 349-residue-long protein involved in regulation of the stability of p21^WAF1/CIP1^ protein, which is a cyclin-dependent kinase inhibitor and a critical regulator of cell cycle, via interaction with Hsp90 and p21^WAF1/CIP1^ [128]. FKBPL contains FKPD-like domain (residues 95–181), and three TPRs (residues 210–243, 252–285, and 286–316; the corresponding ADS values of 0.65 ± 0.15, 0.45 ± 0.13, and 0.41 ± 0.12). This protein contains high levels of intrinsic disorder, with the N-terminal region (residues 1–95) predicted to be completely disordered by all predictors utilized in this study, while the C-terminal region (residues 152-349) is mostly disordered (Figure 22A,B).

Curiously, the FKPD-like domain constitutes the most ordered part of this protein, whereas all three TPRs are located within the mostly disordered C-terminal domain. Figure 22B shows that there are 9 MoRFs in this protein, suggesting high interactability. In line with this indication, Figure 22C represents STRING-generated PPI network centered at FKBPL and shows that this protein can interact with 42 partners. This PPI network includes 120 edges and is characterized by an average node degree of 5.71, average local clustering coefficient of 0.787, expected number of edges of 63, and PPI enrichment *p*-value of < 10^−16^.

Figure 23 reports results of the functional disorder analysis of human kinesin light chain 3 (KLC3; PPDR = 57.2 ± 11.4%; ADS = 0.547 ± 0.070). Kinesin is an oligomer composed of two heavy chains and two light chains that associates with microtubulin in an ATP-dependent manner and serves as a force-producing microtubule-associated protein that may be responsible for organelle transport. The 504-residue-long KLC3 contains a Rab5-binding domain (residues 79–248) and five TPRs (residues 207–240, 249–282, 291–324, 333–366, and 375–408; the corresponding ADS values of 0.30 ± 0.15, 0.25 ± 0.20, 0.44 ± 0.12, 0.25 ± 0.11, and 0.34 ± 0.09). In the N-terminal half of this protein, there is a coiled-coil domain (residues 90–150), as well as Poly-Glu and Poly-Ala regions with compositional bias (residues 181–184 and 190–196, respectively). Figure 23A,B show that 200 N-terminal residues and 118 C-terminal residues (387–504) are predicted to be mostly disordered, and most of the Rab5-binding domain is immerged within the long N-terminal IDPR.

Central TPR-containing domain is characterized by the “wavy” disorder profile, possessing several short IDPRs connecting more ordered regions that mostly correspond to TPRs. Both disordered tails of KLC3 have multiple MoRFs and phosphorylation sites (see Figure 23B). In addition to the canonical form, KLC3 has an alternatively spliced isoform with the methionine at position 1 substituted to MIPQTPHHCSPGAAM.

This addition extends the N-terminal IDPR by 14 residues and extends N-terminal MoRF from 6 to 25 residues. Based on the STRING analysis, KLC3 interacts with 165 proteins, and is located at the center of a dense PPI network (see Figure 23C). This PPI network includes 5,380 edges and is characterized by the average node degree of 64.8, average local clustering coefficient of 0.875, expected number of edges of 456, and PPI enrichment *p*-value of < 10^−16^.

## 3. Materials and Methods

### 3.1. Datasets

On 1 October 2019, all human TPR repeat-containing proteins listed in UniProt (https://www.uniprot.org/) were identified using the search criteria: *tetratricopeptide repeat protein annotation:(type:repeat tpr) AND reviewed:yes AND organism:”Homo sapiens (Human) [9606]”*. This search yielded 166 different human proteins, which were used to perform computational analysis for abundance of intrinsic disorder. Sequences of these proteins in FASTA format and some functional information was extracted from UniProt [129]. UniProt IDs and some related information of the analyzed proteins are shown in the Appendix A. The available protein structures were obtained from the Protein Data Bank (PDB; https://www.rcsb.org/) [130].

### 3.2. Computational Characterization of Intrinsic Disorder in Human TPR Proteins

Initially, human TPR proteins were analyzed by five per-residue predictors, such as PONDR^®^ VLXT [131], PONDR^®^ VSL2 [132], and PONDR^®^ VL3 [132] available on the PONDR site (http://www.pondr.com), and the IUPred computational platform that allows identification of either short or long regions of intrinsic disorder, IUPred-L and IUPred-S [133]. We utilized DiSpi web crawler that was designed for the rapid prediction and comparison of protein disorder profiles. It aggregates the results from a number of well-known disorder predictors: PONDR^®^ VLXT [131], PONDR^®^ VL3 [134], PONDR^®^ VLS2B [132], PONDR^®^ FIT [89], IUPred2 (Short) and IUPred2 (Long) [133,135]. The outputs of the evaluation of the per-residue disorder propensity by these tools are represented as real numbers between 1 (ideal prediction of disorder) and 0 (ideal prediction of order). A threshold of ≥0.5 was used to identify disordered residues and regions in query proteins. For each query protein in this study, the predicted percentage of intrinsic disorder (PPID) was calculated based on the outputs of per-residue disorder predictors. Here, PPID in a query protein represents a percent of residues with disorder scores exceeding 0.5.

Next, the outputs of two binary predictors, the charge-hydropathy (CH) plot [4,14] and the cumulative distribution function (CDF) plot [14,90,136] were combined to conduct a CH-CDF analysis [90,91,92,93] that allows classification of proteins based on their position within the CH-CDF phase space as ordered (proteins predicted to be ordered by both binary predictors), putative native “molten globules” or hybrid proteins (proteins determined to be ordered/compact by CH, but disordered by CDF), putative native coils and native pre-molten globules (proteins predicted to be disordered by both methods), and proteins predicted to be disordered by CH-plot, but ordered by CDF.

Complementary disorder evaluations together with important disorder-related functional information were retrieved from the D^2^P^2^ database (http://d2p2.pro/) [137], which is a database of predicted disorder for a large library of proteins from completely sequenced genomes [137]. D^2^P^2^ database uses outputs of IUPred [133], PONDR^®^ VLXT [131], PrDOS [138], PONDR^®^ VSL2B [134,139], PV2 [137], and ESpritz [140]. The visual console of D^2^P^2^ displays 9 colored bars representing the location of disordered regions as predicted by these different disorder predictors. In the middle of the D^2^P^2^ plots, the blue-green-white bar shows the predicted disorder agreement between nine disorder predictors (IUPred, PONDR^®^ VLXT, PONDR^®^ VSL2, PrDOS, PV2, and ESpritz), with blue and green parts corresponding to disordered regions by consensus. Above the disorder consensus bar are two lines with colored and numbered bars that show the positions of the predicted (mostly structured) SCOP domains [141,142] using the SUPERFAMILY predictor [143]. Yellow zigzagged bar shows the location of the predicted disorder-based binding sites (MoRF regions) identified by the ANCHOR algorithm [144], whereas differently colored circles at the bottom of the plot show location of various PTMs assigned using the outputs of the PhosphoSitePlus platform [145], which is a comprehensive resource of the experimentally determined post-translational modifications.

### 3.3. Computational Evaluation of Interactability of the Human TPR Proteins

Information on the interactability of human TPR proteins was retrieved using Search Tool for the Retrieval of Interacting Genes; STRING, http://string-db.org/. STRING generates a network of protein-protein interactions based on predicted and experimentally-validated information on the interaction partners of a protein of interest [124]. In the corresponding network, the nodes correspond to proteins, whereas the edges show predicted or known functional associations. Seven types of evidence are used to build the corresponding network, where they are indicated by the differently colored lines: a green line represents neighborhood evidence; a red line – the presence of fusion evidence; a purple line – experimental evidence; a blue line – co-occurrence evidence; a light blue line – database evidence; a yellow line – text mining evidence; and a black line – co-expression evidence [124].

In this study, STRING was utilized in three different modes: to generate the inter-TPR network PPI interactions, to produce the TRP set-centered PPI network, and to create PPI networks centered at individual human TPR protein. Protein interaction networks were obtained from STRING (https://string-db.org/) for each protein using a custom value of 500 maximum first-shell interactions at medium, high, and highest confidence levels (minimum required interaction score of 0.4, 0.7, and 0.9 respectively). Resulting PPI networks were further analyzed using STRING-embedded routines in order to retrieve the network-related statistics, such as: the number of nodes (proteins); the number of edges (interactions); average node degree (average number of interactions per protein); average local clustering coefficient (which defines how close the neighbors are to being a complete clique – if a local clustering coefficient is equal to 1, then every neighbor connected to a given node *N_i_* is also connected to every other node within the neighborhood, and if it is equal to 0, then no node that is connected to a given node *N_i_* connects to any other node that is connected to *N_i_*); expected number of edges (which is a number of interactions among the proteins in a random set of proteins of similar size); and a PPI enrichment *p*-value (which is a reflection of the fact that query proteins in the analyzed PPI network have more interactions among themselves than what would be expected for a random set of proteins of similar size, drawn from the genome. It was pointed out that such an enrichment indicates that the proteins are at least partially biologically connected, as a group).

## 4. Conclusions

This article summarizes the results of comprehensive bioinformatics and computational analysis of the intrinsic disorder predisposition of human proteins possessing Tetratricopeptide Repeats (TPRs). This analysis revealed that human TPR proteins, ranging in length from 151 to 4471 residues, contain variable levels of intrinsic disorder (1.9% to 74.9%), with disordered residues assembled into functionally important IDPRs. Disorder in these proteins is used for protein-protein interactions and serves as a signal for a variety of posttranslational modifications. Results of our analysis add a new perspective for better understanding of the functionality of TPR proteins, many of which have high disorder content. They are characterized by the presence of multiple PTM sites, disorder-based protein interaction sites that can undergo binding induced folding at interaction with specific partners, and numerous isoforms generated by alternative splicing. These features (intrinsic disorder, capability to undergo disorder-to-order transitions, PTMs, and alternative splicing) are known to serve as a basis of the proteoform concept, where proteoforms constitute a set of structurally and functionally distinct protein molecules encoded by a single gene [146,147]. They are also a cornerstone of the “protein structure-function continuum” model, where any protein represents a dynamic conformational ensemble, and where multiple proteoforms have different structural features with various functions [146,148,149]. Such disorder-based structural and functional heterogeneity of human TPR proteins can be used to better understanding of the engagement of these proteins in the pathogenesis of various human diseases. This is also in agreement with the observation that IDPs or hybrid proteins containing ordered domains and functional IDPRs are common among disease-associated proteins [45,46]. This is because multi-level harm can be induced by the deregulation, misfolding, misidentification, mis-interactions, and mis-signaling of IDPs/IDPRs, which are commonly found in various proteomes and are involved in recognition, regulation, and cell signaling [45,52,150,151,152,153].

## Figures and Tables

**Figure 1 ijms-21-03709-f001:**
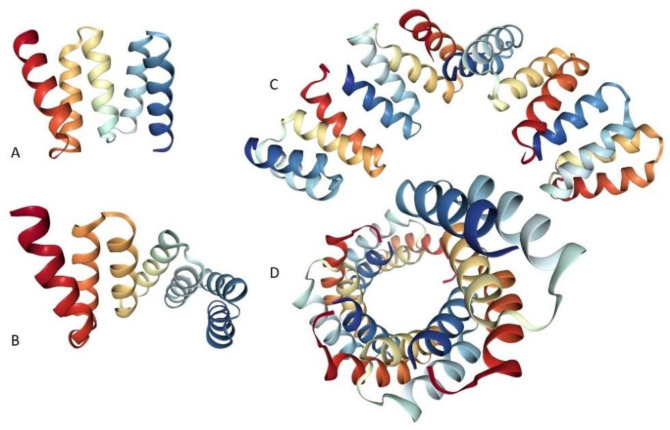
Structural characterization of TPR-containing proteins showing the ability of TPR domains to be stacked continuously one upon the other. Shown here are crystal structures of artificial proteins specially designed to contain two (PDB ID: 1NA3, [72]) (**A**), three (PDB ID: 1NA3, [72]) (**B**), or 20 copies of the consensus TPR motif (**C** and **D**) (PDB ID: 2AVP, [73]), i.e., CTPR2, CTPR3, and CTPR20, respectively. To emphasize the ability of TPR proteins to form specific superhelical structure, plots **C** and **D** show two projections (side view and top view, respectively) of CTPR20.

**Figure 2 ijms-21-03709-f002:**
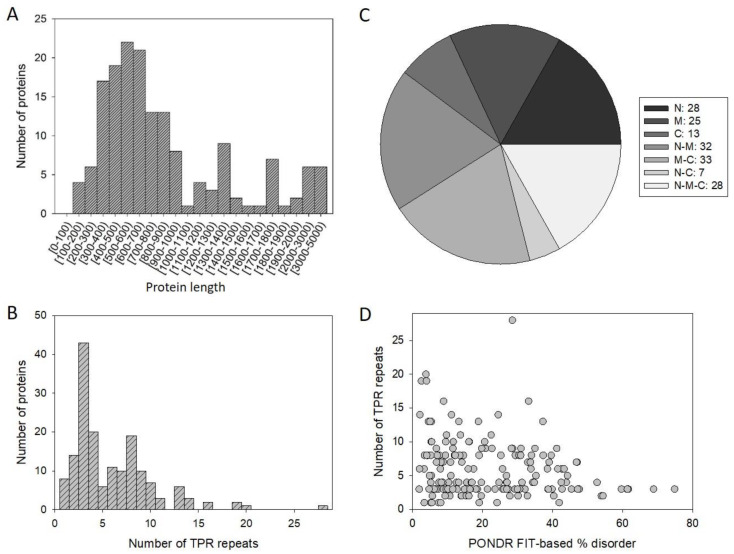
Some characteristics of 166 human TPR proteins analyzed in this study. (**A**) Length distribution of studied proteins. (**B**) Quantification of the number of TPR repeats in the individual proteins. (**C**) Quantification of the peculiarities of TPR repeat distribution within the amino acid sequences of human TPR proteins: N, M, C, N-M, M-C, N-C, and N-M-C correspond to the preferential potion of TPR repeats within N-terminal (N), middle (M), or C-terminal (C) parts of the sequence and their various combinations. (**D**) Correlation between the number of TPR repeat in query proteins and their intrinsic disorder status.

**Figure 3 ijms-21-03709-f003:**
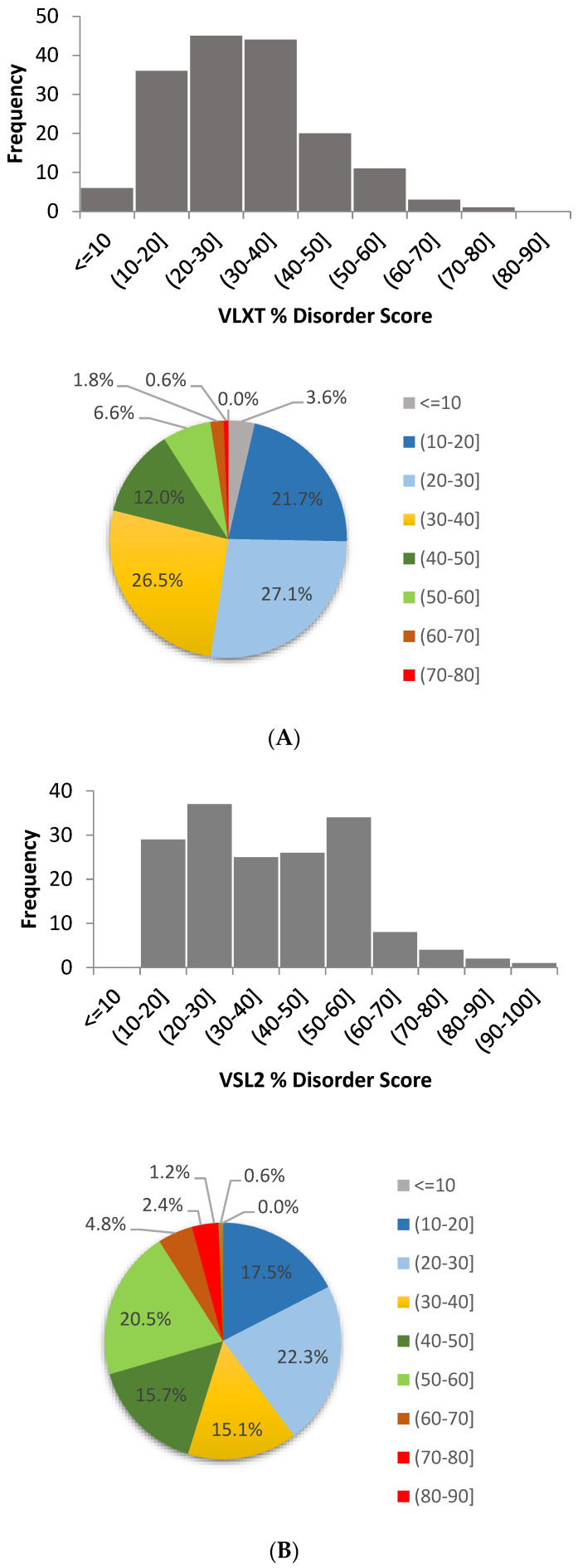
Frequency distributions of the percent of predicted disorder in 166 human TPR proteins as evaluated by PONDR^®^ VLXT (**A**), PONDR^®^ VSL2 (**B**), and PONDR^®^ FIT (**C**). (**A**) Frequency distribution of predicted percent of disorder evaluated by PONDR^®^ VLXT; (**B**) Frequency distribution of predicted percent of disorder evaluated by PONDR^®^ VSL2; (**C**) Frequency distribution of predicted percent of disorder evaluated by PONDR^®^ FIT.

**Figure 4 ijms-21-03709-f004:**
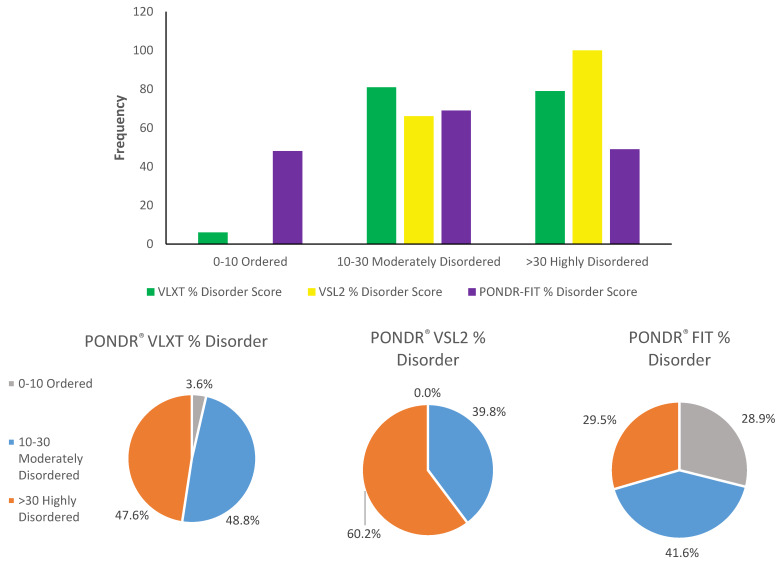
Distribution of proteins based on percent disorder score from PONDR^®^ VLXT, PONDR^®^ VSL2, and PONDR^®^ FIT. The category 0%–10% are considered highly ordered proteins, 10%–30% are moderately disordered, and greater than 30% are highly disordered.

**Figure 5 ijms-21-03709-f005:**
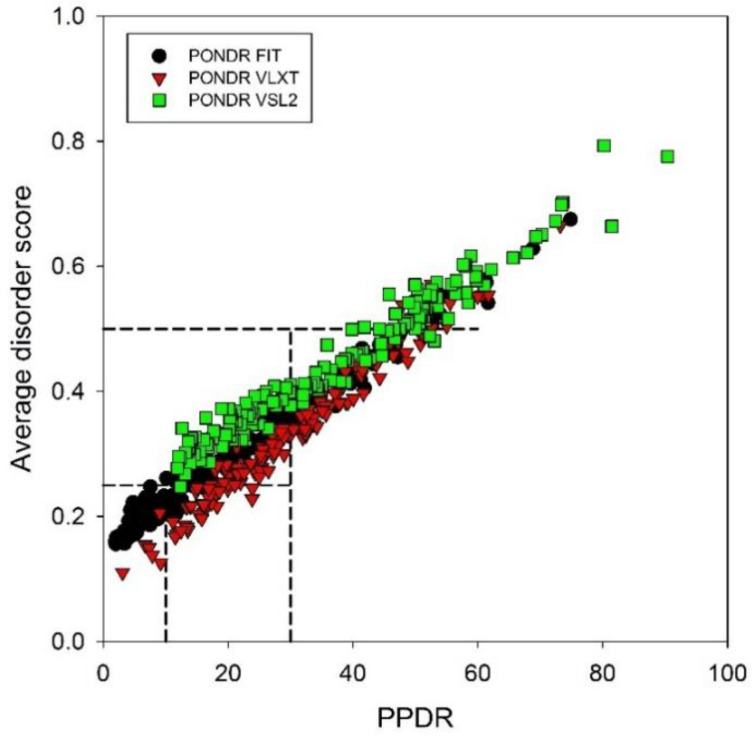
Plots correlating the average disorder scores and PPDRs for human TPR proteins based on the results of PONDR^®^ VLXT, PONDR^®^ VSL2, and PONDR^®^ FIT analyses. Dashed lines represent boundaries separating ordered (ADS < 0.25, PPDR < 10%), moderately disordered (0.25 ≤ ADS < 0.5, 10% ≤ PPDR < 30%), and highly disordered proteins (ADS ≥ 0.5, PPDR ≥ 30%).

**Figure 6 ijms-21-03709-f006:**
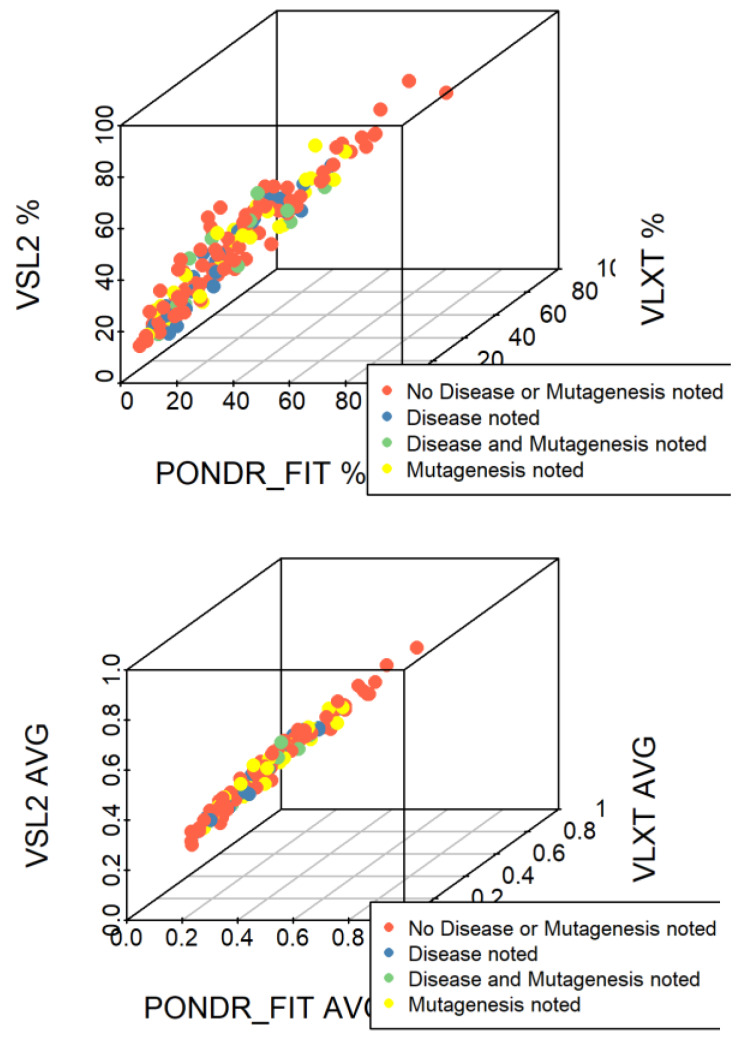
Three-dimensional scatterplots comparing the average disorder scores and average PPDR values generated by three different predictors. On the left panel: average disorder scores generated by PONDR^®^ VLXT, PONDR^®^ VSL2, and PONDR^®^ FIT along the x, y, and z axes, respectively. On the right panel: PPDR evaluated by PONDR^®^ VLXT, PONDR^®^ VSL2, and PONDR^®^ FIT along the x, y, and z axes, respectively. Proteins shown in red had no associated disease or mutagenesis information in UniProt, while those shown in green had both associated disease and mutagenesis information. Those in blue had only associated disease information, and those and those in yellow had only associated mutagenesis information.

**Figure 7 ijms-21-03709-f007:**
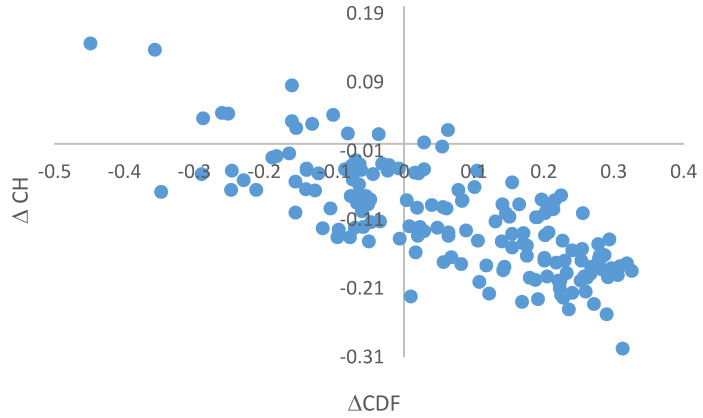
CH-CDF plot analysis of 166 human TPR proteins. Quadrants are numbered in a clockwise direction starting with Q1 in the lower right and ending with Q4 in the upper right. The following characterize proteins within each quadrant: Q1 (lower-right quadrant) – proteins predicted to be ordered by both predictors, Q2 (lower-left quadrant) – proteins predicted to be ordered by CH but disordered by CDF, Q3 (upper-left quadrant) – proteins predicted to be disordered by both predictors, and Q4 (upper-right quadrant) – proteins predicted to be disordered by CH and ordered by CDF.

**Figure 8 ijms-21-03709-f008:**
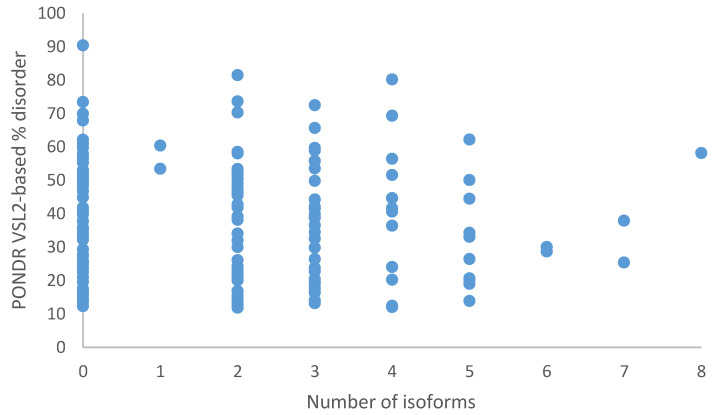
The number of protein isoforms listed in UniProt against the PONDR^®^ VSL2 percent disorder score.

**Figure 9 ijms-21-03709-f009:**
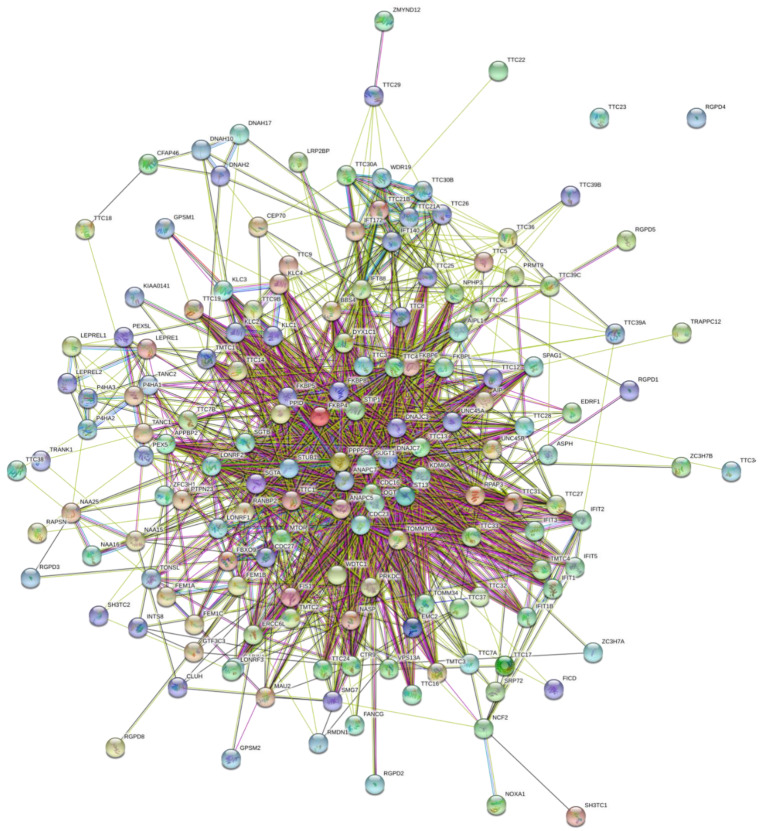
STRING-based analysis of the inter-set interactivity of 166 human TPR proteins using the low confidence level of 0.15. This confidence level was selected to ensure maximal inclusion of TPR proteins in the resulting PPI. STRING generates a network of predicted associations based on predicted and experimentally-validated information on the interaction partners of a protein of interest [124]. In the corresponding network, the nodes correspond to proteins, whereas the edges show predicted or known functional associations. Seven types of evidence are used to build the corresponding network, and are indicated by the differently colored lines: a green line represents neighborhood evidence; a red line – the presence of fusion evidence; a purple line – experimental evidence; a blue line – co-occurrence evidence; a light blue line – database evidence; a yellow line – text mining evidence; and a black line – co-expression evidence [124].

**Figure 10 ijms-21-03709-f010:**
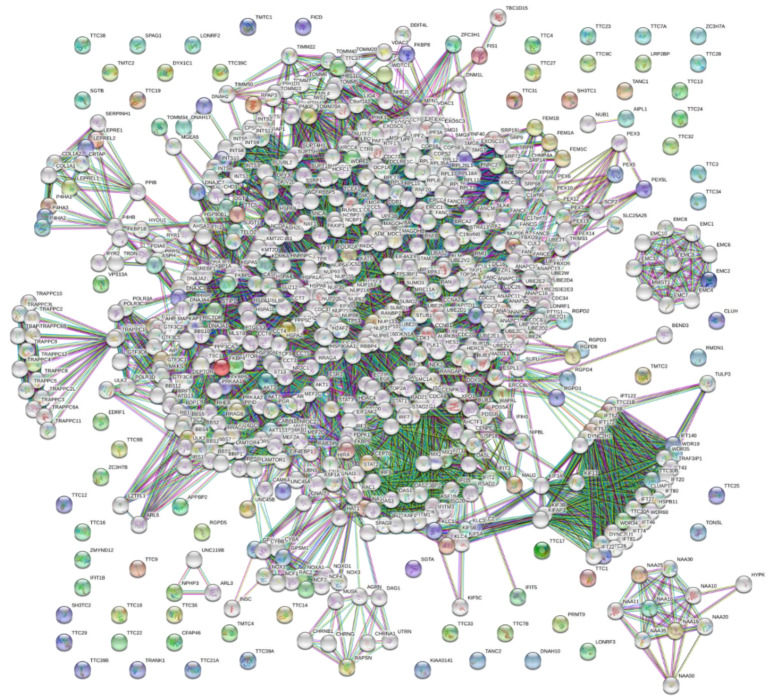
STRING-based analysis of the whole set of human TPR proteins using the highest confidence level of 0.9. This confidence level was selected to ensure maximal inclusion of TPR proteins in the resulting PPI.

**Figure 11 ijms-21-03709-f011:**
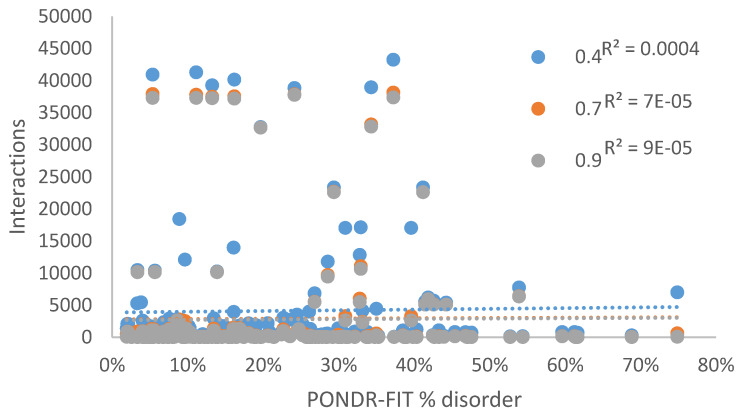
The quantity of protein interactions given by STRING at medium (0.4), high (0.7), and highest (0.9) confidence by the PONDR-FIT percent disorder score.

**Figure 12 ijms-21-03709-f012:**
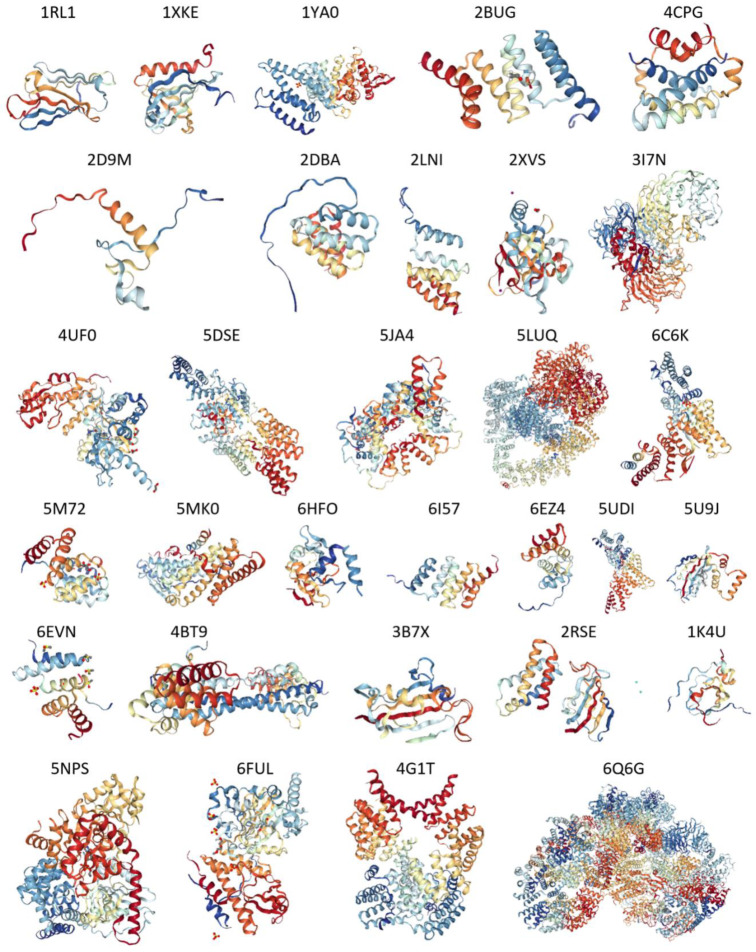
A portrait gallery of 56 human TPR proteins with structure available in PDB. If multiple structures were available for a given protein, the structure with the greatest number of residues was selected, otherwise, the X-ray crystallographic structure with the best resolution was selected. For proteins with NMR structures, only one model is shown.

**Figure 13 ijms-21-03709-f013:**
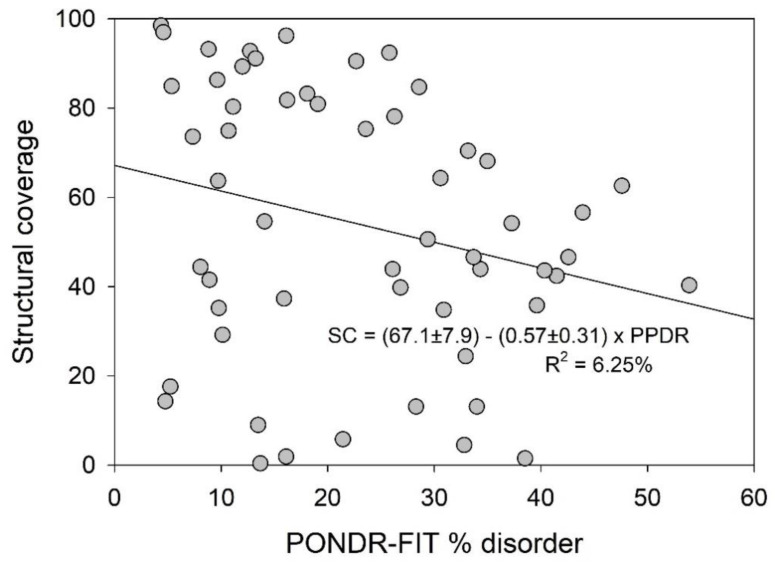
Correlation of the structural coverage (percent of protein sequence that map to known structures) of 56 human TPR proteins with their intrinsic disorder content evaluated as PONDR^®^ FIT-based PPDR values.

**Figure 14 ijms-21-03709-f014:**
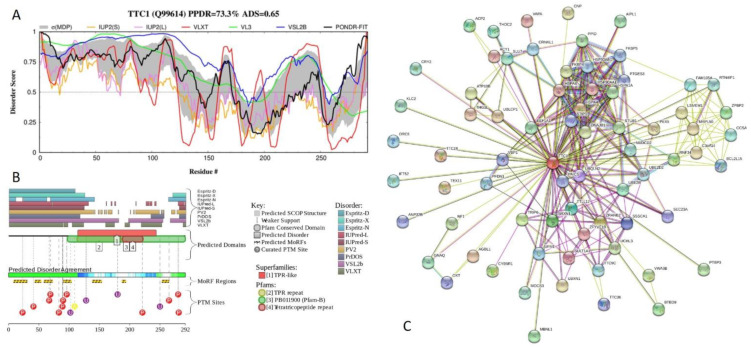
Functional disorder analysis of human tetratricopeptide repeat protein 1 (TTC1; UniProt ID: Q99614). (**A**) Intrinsic disorder profile generated by DiSpi web crawler. (**B**) Functional disorder profile generated by D^2^P^2^ platform. (**C**) Interactability of human TTC1 protein analyzed by STRING.

**Figure 15 ijms-21-03709-f015:**
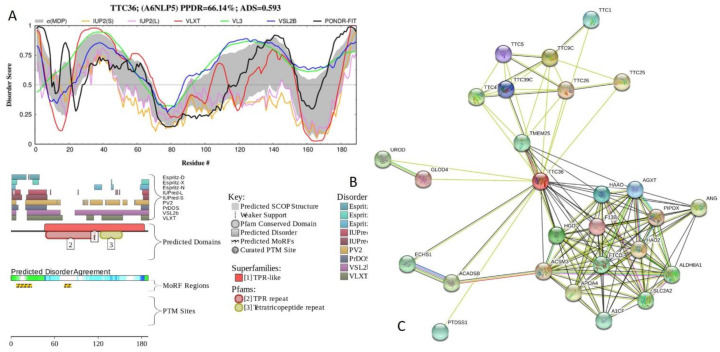
Functional disorder analysis of human tetratricopeptide repeat protein 36 (TTC36; UniProt ID: A6NLP5). (**A**) Intrinsic disorder profile generated by DiSpi web crawler. (**B**) Functional disorder profile generated by D^2^P^2^ platform. (**C**) Interactability of human TTC36 protein analyzed by STRING.

**Figure 16 ijms-21-03709-f016:**
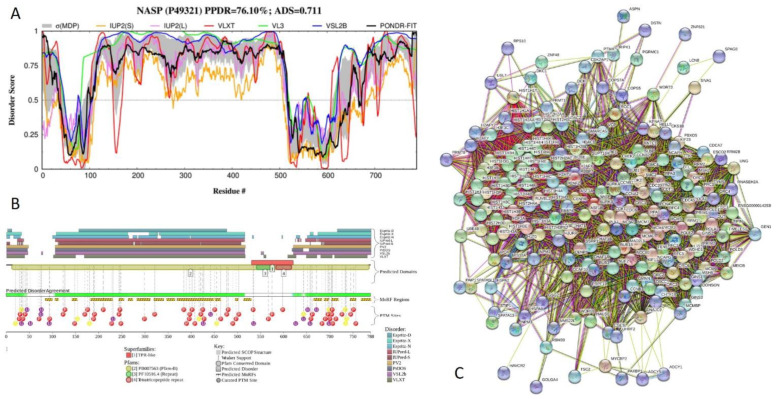
Functional disorder analysis of human nuclear autoantigenic sperm protein (NASP; UniProt ID: P49321). (**A**) Intrinsic disorder profile generated by DiSpi web crawler. (**B**) Functional disorder profile generated by D^2^P^2^ platform. (**C**) Interactability of human NASP analyzed by STRING.

**Figure 17 ijms-21-03709-f017:**
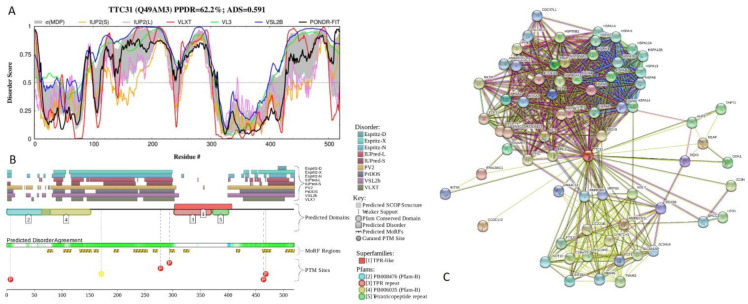
Functional disorder analysis of human tetratricopeptide repeat protein 31 (TTC31; UniProt ID: Q49AM3). (**A**) Intrinsic disorder profile generated by DiSpi web crawler. (**B**) Functional disorder profile generated by D^2^P^2^ platform. (**C**) Interactability of human TTC31 protein analyzed by STRING using the medium confidence level of 0.4.

**Figure 18 ijms-21-03709-f018:**
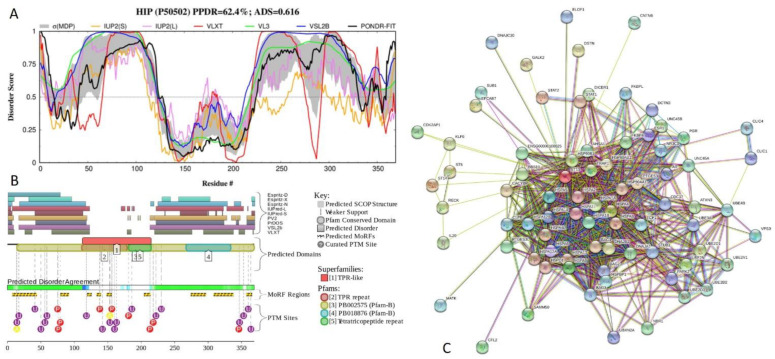
Functional disorder analysis of human Hsc70-interacting protein (HIP, or protein FAM10A1; UniProt ID: P50502). (**A**) Intrinsic disorder profile generated by DiSpi web crawler. (**B**) Functional disorder profile generated by D^2^P^2^ platform. (**C**) Interactability of human HIP analyzed by STRING.

**Figure 19 ijms-21-03709-f019:**
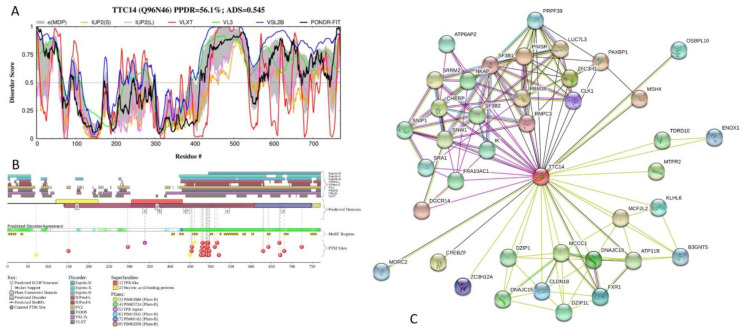
Functional disorder analysis of human tetratricopeptide repeat protein 14 (TTC14; UniProt ID: Q96N46). (**A**) Intrinsic disorder profile generated by DiSpi web crawler. (**B**) Functional disorder profile generated by D^2^P^2^ platform. (**C**) Interactability of human TTC14 protein analyzed by STRING using medium confidence level of 0.4.

**Figure 20 ijms-21-03709-f020:**
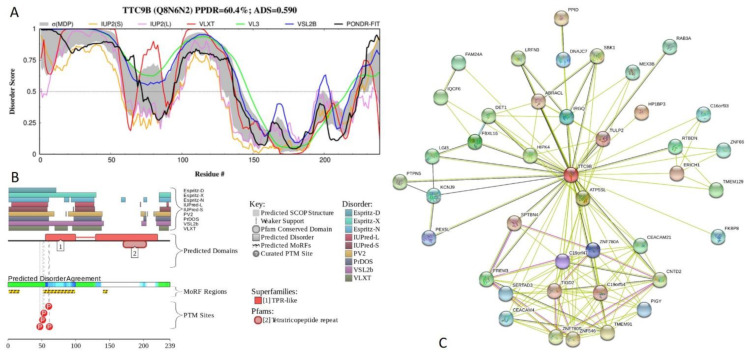
Functional disorder analysis of human tetratricopeptide repeat protein 9B (TTC9B; UniProt ID: Q8N6N2). (**A**) Intrinsic disorder profile generated by DiSpi web crawler. (**B**) Functional disorder profile generated by D^2^P^2^ platform. (**C**) Interactability of human TTC9B protein analyzed by STRING using the medium confidence level of 0.4.

**Figure 21 ijms-21-03709-f021:**
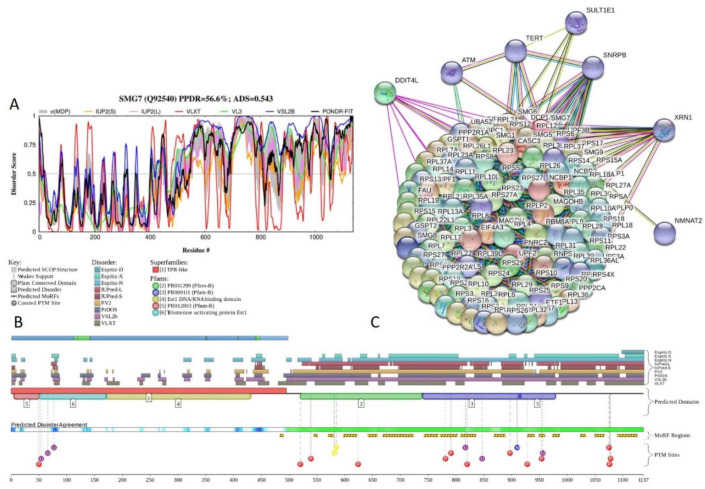
Functional disorder analysis of human protein SMG7 (UniProt ID: Q92540). (**A**) Intrinsic disorder profile generated by DiSpi web crawler. (**B**) Functional disorder profile generated by D^2^P^2^ platform. (**C**) Interactability of human SMG7 protein analyzed by STRING using the high confidence level of 0.7.

**Figure 22 ijms-21-03709-f022:**
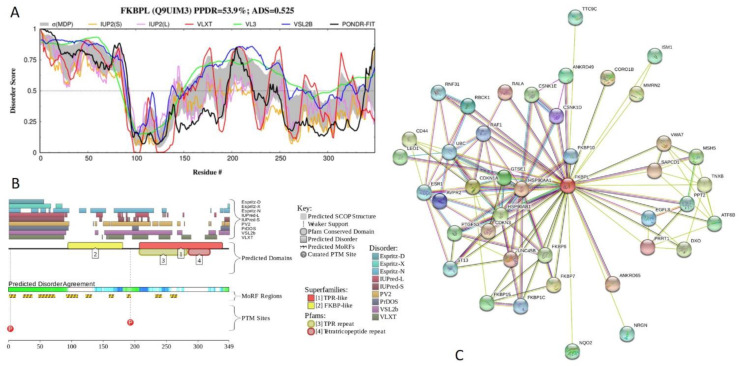
Functional disorder analysis of human FK506-binding protein-like (FKBPL; UniProt ID: Q9UIM3). (**A**) Intrinsic disorder profile generated by DiSpi web crawler. (**B**) Functional disorder profile generated by D^2^P^2^ platform. (**C**) Interactability of human FKBPL protein analyzed by STRING using the medium confidence level of 0.4.

**Figure 23 ijms-21-03709-f023:**
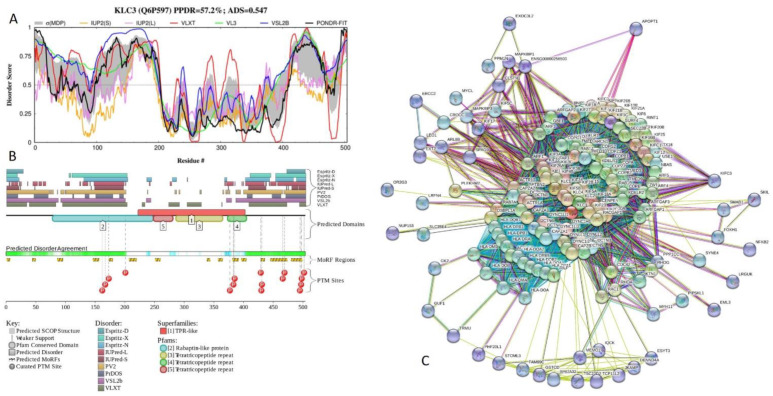
Functional disorder analysis of human kinesin light chain 3 (KLC3; UniProt ID: Q6P597). (**A**) Intrinsic disorder profile generated by DiSpi web crawler. (**B**) Functional disorder profile generated by D^2^P^2^ platform. (**C**) Interactability of human KLC3 analyzed by STRING using the medium confidence level of 0.4.

**Table 1 ijms-21-03709-t001:** Distribution of the number of disordered regions (# DR) and longest disordered region (Longest DR) for the PONDR^®^ VLXT and PONDR^®^ VSL2 predictors.

	PONDR^®^ VLXT		PONDR^®^ VSL2	
	# DR	Longest DR	# DR	Longest DR
Min	3	10	3	11
Average	14.5	61.2	14.0	128.5
Max	61	348	68	818

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
