# Peer review of "Intrinsic Disorder in Tetratricopeptide Repeat Proteins"

_ijms, 2020, doi:10.3390/ijms21103709_

Round 1

Reviewer 1 Report

The manuscript by Van Bibber and coworkers is a bioinformatics study of the Tetratricopeptide Repeat Proteins (TRP) to characterize the extent of intrinsic disorder within the protein family.  The authors use amino acid sequence analysis to estimate regions of intrinsic disorder along with database analysis of posttranslational modifications, disease mutations and protein interactions to anecdotally examine function.  These are an important and diverse proteins that are linked to multiprotein complex formation (i.e. scaffolding). Understanding disorder within the protein family is important as disorder has been linked to protein interactions and scaffolding.  The manuscript is well written and clear but the introduction could be more focused on TPRs rather than IDPs generally. While there is little issue with any of the data and analysis, the paper is very short on conclusions and interference based on the analysis. Given the amount of work presented, it should be possible to present more meta-analysis to draw general conclusions about the TPR protein family rather than just present anecdotal examples. Differences in the disorder predictors seem significant but are not well discussed. I was confused about the origin and utility of making a binary (yes/no) assignment to a protein being disordered. Clearly, the individual examples are more complex than this simple number. The authors should be able to provide a more nuanced description of the context of TPR repeats. I also did not see the value in the STRING analysis, which is not used to make any conclusions. In general, the whole paper would benefit from a stronger conclusion about disorder in the TPR family. There is a lot of good work here and it is my belief that the authors can address these points in revision.

MAJOR COMMENTS

The paper contains much tangential information while some key concepts could use more explanation. The introduction is written at a very general level with some historical review that does not well serve introducing the research project at hand. I recommend cutting everything up to Page 3 line 17 and begin by discussing scaffolding, tandem repeats and the structure/function of natural TPR proteins. These topics could use more introduction to establish the importance of the research presented.  I would suggest focusing more on the biological diversity of natural TPR proteins since exploring natural sequence diversity is the topic of research. Along these lines, Figure 1 should show the structure of natural rather than artificial proteins. Additionally, page 11 contains a discourse on IDP interactions and their fuzziness.  TPR repeats are structured and would be considered a peptide binding motif. Thus, this discussion does not appear to be relevant to TPR interactions so the inclusion is not clear. While low affinity, individual TPR proteins are not “morphing shape changers.”  This discussion should be limited and focus on the TPR repeats instead of general principal of IDPs. TPR motifs are known to appear in scaffolding proteins so their multivalency has been established.

The different predictors are giving incompatible estimates of the extent of disorder. In particular, FIT predicts a high occurrence of ordered TPR proteins (29%). The differences between predictors are not entirely clear nor is the rationale for their inclusion.  Should one be better at this than another? These differences are not discussed sufficiently.  Can these predictions be checked (at least in a couple instances) by looking for known folded domains or some other means to identify folded regions in these genes? It is stated that Figure 6 indicates good agreement between the disorder predictors.  This is confusing since Figures 3 and 4 show the pronounced differences between them in classifying proteins into three classes of disorder. I am unclear what this representation is showing.

I am not clear the utility of assigning a protein a mean disorder percentage. The presence of highly disordered regions within TPR proteins would be obscured by such averaging. At different points in the text, criteria are explained for categorizing IDPs based on mean % disorder. On page 6, two different cutoff ranges are given for ADS and PPDR, which raises confusion. The origin of these cutoff values is not clear. A reference could be provided. I am unclear why 30% disorder is highly disordered.  By this rationale, a 100 residue protein containing a folded domain of 70aa with disordered termini of 15aa on each end would be considered highly disordered even though it is mostly folded.

The disorder predictors use sequence complexity as an indicator of disorder. To what extent does the presence of TPR repeats make these TPR proteins low complexity in this analysis? Are the TRP repeats excluded from the disorder analysis? To what extent are the TPR predicted to be ordered vs. disordered?

Can some general conclusions be made about the context of TPR repeats?  Are they more likely to be N- or C-terminal? Are they likely to be paired with any other known motifs or folded domains?  Does the number of TPR correlated with the extent disorder? Can the author show a distribution of the number of TPR repeats in the dataset? What is distribution and mean length of the IDRs in these proteins?  The text refers to Figure 3A as showing this data but I am not finding it.

The conclusions from the STRING analysis throughout the text are unclear.  This is a graphical representation of publically available data for known and predicted interactions for each protein. Not all of these proteins may be expressed in the same cells at the same time.  Some proteins are from prokaryotes? How many predicted interactions are compatible with biological expression data? What fraction of nodes are predicted vs. experimentally confirmed? IsI don’t see information being extracted from the pattern of connectivity and there is little metaanalysis of the entire dataset.  Figure 11 shows that connectivity does not correlate so this seems off topic for a paper about disorder in TPR proteins. I recommend removal of these analyses or addition of some meta-analysis to draw conclusions about the protein family. Additionally, these figures are too small to be legible.

The purpose of showing the large number of structures in Figure 12 is not clear. It was known a priori that TPR repeats are helical so the presence of helicity is not surprising.  Furthermore, the structures shown are not representing entire proteins but rather the crystalizable domains from these proteins.  The analysis herein shows that many if not most TPR repeats are associated with disorder.  What fraction of the gene is represented by the ordered domains shown? I am not clear what is to be taken from visual inspection of 50+ published structures without any associated analysis?  If the figure does not support any conclusions it should be modified or removed.

The D2p2 analysis not well described in the text. There is much more information shown in the figures beyond PTMs, which is all that is indicated in the text. These are all very hard to read as the text is too small to be legible on a printed page. More details are needed about the analysis and a better legend is needed to explain the extensive number of colors and annotations. Where is the data for PTMs coming from in the D2P2 analysis?  I tired checking some of these and did not find that data in the database.

The rationale for presenting 10 individual anecdotal protein examples is not clear and substantially increases the number of figures in the main text (Figures 13-22). . Is there some significance to the top 10 % disordered proteins or why was this number chosen? I believe that these figures could be in the Supporting Information. There should be some meta-analysis or conclusions drawn from these analyses.  

It is stated in the abstract that PTMs in TPR proteins are associated with the IDRs.  Is there a global analysis that supports this or is this coming from the anecdotal analysis of the 10 most disordered proteins? I see some PTMs in the order regions of the D2P2 analysis.

MINOR COMMENTS

I am unclear what “reactions” are affected by TRP repeat proteins.  Is this referring to protein binding as a reaction or is there some enzymatic reactions being regulated?

I am not clear what the definition here is for “stacking” of TPR repeats since the repeats are offset by a helical twist. All tandem repeats follow each other with some contact.  Is this aromatic stacking or do all tandem repeats stack?

I am failing to appreciate the remarkableness of TPR designability and engineering.  There are many protein scaffolds that have been engineered for alternate specificity and higher affinity. Why does it matter that these repeats can be concatenated unnaturally without limit?  I would suggest toning down language about exceptionality unless directly justified in the text.

What is an absolute majority?

MoRFs are not well introduced before they are analyzed. I am unclear how these are being predicted in the current work.

TYPOS

The Figure 1 legend should read “…ability [of] TPR domains to be stack continuously one upon the other.”

The colors in Figure 6  are not in agreement with the description in the main text.

The discussion of Figure 13 is jumbled.  The text states that Figure 13A identifies the location of TPR repeats in the N-termnus but the numbers quoted are in the middle of the protein.  The sentence appears to be missing a clause as it ends with “and.”

Author Response

The manuscript by Van Bibber and coworkers is a bioinformatics study of the Tetratricopeptide Repeat Proteins (TPR) to characterize the extent of intrinsic disorder within the protein family.  The authors use amino acid sequence analysis to estimate regions of intrinsic disorder along with database analysis of posttranslational modifications, disease mutations and protein interactions to anecdotally examine function.  These are an important and diverse proteins that are linked to multiprotein complex formation (i.e. scaffolding). Understanding disorder within the protein family is important as disorder has been linked to protein interactions and scaffolding.  The manuscript is well written and clear but the introduction could be more focused on TPRs rather than IDPs generally. While there is little issue with any of the data and analysis, the paper is very short on conclusions and interference based on the analysis. Given the amount of work presented, it should be possible to present more meta-analysis to draw general conclusions about the TPR protein family rather than just present anecdotal examples. Differences in the disorder predictors seem significant but are not well discussed. I was confused about the origin and utility of making a binary (yes/no) assignment to a protein being disordered. Clearly, the individual examples are more complex than this simple number. The authors should be able to provide a more nuanced description of the context of TPR repeats. I also did not see the value in the STRING analysis, which is not used to make any conclusions. In general, the whole paper would benefit from a stronger conclusion about disorder in the TPR family. There is a lot of good work here and it is my belief that the authors can address these points in revision.

REPLY: We are thankful to this reviewer for careful reading of the manuscript, high evaluation of our work and constructive criticism. We tried to address all the queries and hope that the revised manuscript became more suitable for publication.

MAJOR COMMENTS

  1. The paper contains much tangential information while some key concepts could use more explanation. The introduction is written at a very general level with some historical review that does not well serve introducing the research project at hand. I recommend cutting everything up to Page 3 line 17.

REPLY: We are thankful to this reviewer for pointing this out. Although the reviewer is right, and the introduction is written at a very general level, we do believe that this introduction is needed, since not everybody in the field is familiar with the intrinsic disorder phenomenon. Therefore, we decided to keep this part in the manuscript.

  • … and begin by discussing scaffolding, tandem repeats and the structure/function of natural TPR proteins. These topics could use more introduction to establish the importance of the research presented. I would suggest focusing more on the biological diversity of natural TPR proteins since exploring natural sequence diversity is the topic of research. Along these lines, Figure 1 should show the structure of natural rather than artificial proteins.

REPLY: We are thankful to this reviewer for pointing this out. We added better introduction to the scaffolding and tandem repeats. We also included a paragraph on the structure/function of natural TPR proteins to better emphasize the biological diversity of natural TPR proteins. Since a whole set of structures of natural TPRs is shown in Figure 12, and since Figure 1 provides a nice outlook of the stackability and designability of TPR proteins, we decided to keep this Figure 1 unaltered.

  • Additionally, page 11 contains a discourse on IDP interactions and their fuzziness. TPR repeats are structured and would be considered a peptide binding motif. Thus, this discussion does not appear to be relevant to TPR interactions so the inclusion is not clear. While low affinity, individual TPR proteins are not “morphing shape changers.”  This discussion should be limited and focus on the TPR repeats instead of general principal of IDPs. TPR motifs are known to appear in scaffolding proteins so their multivalency has been established.

REPLY: We are thankful to this reviewer for pointing this out. Although the reviewer is right, and TPR motifs are typically ordered, many TPR proteins contain IDPRs that alse can be involved in protein-protein interactions and, therefore, are the “morphing shape changers.” Because of this fact, we are not sure that it would be right to focus only at the TPR repeats and not to consider the TPR proteins in their entirety. The (more correct) general picture can be obtained by looking at the TPR repeats and other regions.  Corresponding discussion is added to the revised manuscript.  

  1. The different predictors are giving incompatible estimates of the extent of disorder. In particular, FIT predicts a high occurrence of ordered TPR proteins (29%). The differences between predictors are not entirely clear nor is the rationale for their inclusion. Should one be better at this than another? These differences are not discussed sufficiently. 

REPLY: We are thankful to this reviewer for pointing this out. The use multiple computational tools for prediction of intrinsic disorder in proteins is an accepted practice in the field. Since different computational tools use different attributes and models for finding disordered residues and regions, it is a common situation, when different tools will generate rather different outputs. There is no an accepted consensus of which disorder predictor is the best in evaluating disorder predisposition of proteins. In reality, different tools are sensitive to different disorder-related aspects. In reality, different tools are sensitive to different disorder-related aspects and therefore all of them contain some useful information.  Corresponding clarification is added to the revised manuscript.

  • Can these predictions be checked (at least in a couple instances) by looking for known folded domains or some other means to identify folded regions in these genes?

REPLY: We are thankful to this reviewer for pointing this out. Conducting the proposed experiment (checking predictors by looking for known folded domains or some other means to identify folded regions in these proteins) is outside the scopes of this study.

  • It is stated that Figure 6 indicates good agreement between the disorder predictors. This is confusing since Figures 3 and 4 show the pronounced differences between them in classifying proteins into three classes of disorder. I am unclear what this representation is showing.

REPLY: We are thankful to this reviewer for pointing this out. The fact that the majority of points corresponding to human TPR proteins are located either on or in the close proximity to a diagonal in the corresponding 3D plots in Figure 6 is a reflection of the fact that the outputs of PONDR® VLXT, PONDR® VSL2, and PONDR® FIT for 166 human TPR proteins are generally agree with each other. Obviously, they are not identical and this is reflected in the noticed differences between the tools in classifying proteins into three classes of disorder.

  1. I am not clear the utility of assigning a protein a mean disorder percentage. The presence of highly disordered regions within TPR proteins would be obscured by such averaging. At different points in the text, criteria are explained for categorizing IDPs based on mean % disorder. On page 6, two different cutoff ranges are given for ADS and PPDR, which raises confusion. The origin of these cutoff values is not clear. A reference could be provided. I am unclear why 30% disorder is highly disordered. By this rationale, a 100 residue protein containing a folded domain of 70aa with disordered termini of 15aa on each end would be considered highly disordered even though it is mostly folded.

REPLY: We are thankful to the reviewer for pointing this out. ADS and PPDR were introduced for the quantification of disorder status of large protein sets. Although these numbers could be not too informative for a given protein, they do show a general picture for large datasets. As it is indicated in the text explaining Figure 5, these two measures are not identical, since, theoretically, a protein with the PPDR of 100% might have the ADS ranging from 0.5 to 1.0; whereas a protein with the PPDR of 0% might have any ADS < 0.5. The cutoff ADS and PPDR values for classification of proteins as highly ordered, moderately and highly disordered are arbitrary. For PPDR, the cutoff values were proposed almost a decade ago (Rajagopalan, K.; Mooney, S.M.; Parekh, N.; Getzenberg, R.H.; Kulkarni, P. A majority of the cancer/testis antigens are intrinsically disordered proteins. J Cell Biochem 2011, 112, 3256-3267, doi:10.1002/jcb.23252, which is our reference #73) and are rather intensively used in the literature since then. I think that the perception of the discussed case is subjective and depends on the personal viewpoint. In fact, the order-centric person would focus on the fact that 70 residues are mostly ordered whereas the disorder-centric person would be excited by the presence of 30 disordered residues.

  1. The disorder predictors use sequence complexity as an indicator of disorder. To what extent does the presence of TPR repeats make these TPR proteins low complexity in this analysis? Are the TRP repeats excluded from the disorder analysis? To what extent are the TPR predicted to be ordered vs. disordered?

REPLY: We are thankful to the reviewer for pointing this out. Majority of disorder predictors use amino acid composition as one of the important attributes. Amino acid compositions of TPR motifs are not too biased and definitely are not characterized by low local complexity. In fact, they do not have too many local repeats. Therefore, the presence of TPR repeats does not make TPR proteins low complexity. The TPR repeats are not excluded from the disorder analysis. In these analyses, the TPR motifs are predicted to be (dis)ordered to different degree. This is illustrated by providing ADS values for 10 most disordered TPR proteins discussed in the manuscript. 

  1. Can some general conclusions be made about the context of TPR repeats? Are they more likely to be N- or C-terminal? Are they likely to be paired with any other known motifs or folded domains?  Does the number of TPR correlated with the extent disorder? Can the author show a distribution of the number of TPR repeats in the dataset? What is distribution and mean length of the IDRs in these proteins?  The text refers to Figure 3A as showing this data but I am not finding it.

REPLY:  We are thankful to the reviewer for pointing this out. The recommended analyses are added to the revised manuscript in a form of new plots in Figure 2, as well as textual description of other functional domains an motifs found in human TPR proteins. We also added table 1 to include information on the distribution of the number of disordered regions and longest disorder region.

  1. The conclusions from the STRING analysis throughout the text are unclear. This is a graphical representation of publically available data for known and predicted interactions for each protein. Not all of these proteins may be expressed in the same cells at the same time.  Some proteins are from prokaryotes? How many predicted interactions are compatible with biological expression data? What fraction of nodes are predicted vs. experimentally confirmed? I don’t see information being extracted from the pattern of connectivity and there is little metaanalysis of the entire dataset.  Figure 11 shows that connectivity does not correlate so this seems off topic for a paper about disorder in TPR proteins. I recommend removal of these analyses or addition of some meta-analysis to draw conclusions about the protein family. Additionally, these figures are too small to be legible.

REPLY:  We are thankful to the reviewer for pointing this out. The overall idea of using STRING analysis is to show the high level of interactability of TPR proteins. Although the reviewer is right and not all these interacting proteins may be expressed in the same cell at the same time. However, all of them are experimentally validated or at least predicted as binding partners of corresponding TPR proteins. Therefore, in our view, the corresponding plots should be present in the figures discussing disorder and functionality of highly disordered TPR proteins. As far as we know, prokaryotic proteins can be potentially present in the STRING-generated PPI network, if these proteins were actually shown to be involved in interaction with query proteins. We think that the robust and systematic meta-analysis of the entire dataset, a well as validation of such analysis in a form of the investigation of the compatibility of predicted interactions with biological expression data and evaluation of a fraction of nodes that are predicted vs. experimentally confirmed are outside the scopes of this article and represent a subject of a separate study. This is because such new study would represent a form of validation of the accuracy of STRING, which is definitely the task for the team developed this tool. Although Figure 11 shows that connectivity does not correlate with the intrinsic disorder level, we still think that this is useful information that should be present in the manuscript. 

  1. The purpose of showing the large number of structures in Figure 12 is not clear. It was known a priori that TPR repeats are helical so the presence of helicity is not surprising. Furthermore, the structures shown are not representing entire proteins but rather the crystalizable domains from these proteins.  The analysis herein shows that many if not most TPR repeats are associated with disorder.  What fraction of the gene is represented by the ordered domains shown? I am not clear what is to be taken from visual inspection of 50+ published structures without any associated analysis?  If the figure does not support any conclusions it should be modified or removed.

REPLY:  We are thankful to the reviewer for pointing this out. In our view, the purpose of showing the large number of structures in Figure 12 is clear. Although the reviewer is right and it was known a priori that TPR repeats are helical, Figure 12 indicates than not all human TPR proteins are helical. We completely agree that structures shown in this figure are not representing entire proteins but rather the crystalizable domains from these proteins. Furthermore, some of the structures are not the TPR motifs of the corresponding proteins but their other functional domains. We believe that such a figure should be present, since it gives a nice overview of the ordered side of human TPR proteins. We conducted suggested analysis on the fraction of the protein represented by the experimentally validated ordered domains. Results of this analysis are now shown in new Figure 13.

  1. The D2p2 analysis not well described in the text. There is much more information shown in the figures beyond PTMs, which is all that is indicated in the text. These are all very hard to read as the text is too small to be legible on a printed page. More details are needed about the analysis and a better legend is needed to explain the extensive number of colors and annotations. Where is the data for PTMs coming from in the D2P2 analysis? I tired checking some of these and did not find that data in the database.

REPLY:  We are thankful to the reviewer for pointing this out. Better description of the D2P2 analysis is now provided in the Materials and Method section. The data for PTMs are coming from the outputs of the PhosphoSitePlus platform, which is a comprehensive resource of the experimentally determined post-translational modifications. This information is also included in Materials and Methods.

  1. The rationale for presenting 10 individual anecdotal protein examples is not clear and substantially increases the number of figures in the main text (Figures 13-22). . Is there some significance to the top 10 % disordered proteins or why was this number chosen? I believe that these figures could be in the Supporting Information. There should be some meta-analysis or conclusions drawn from these analyses.

REPLY:  We are thankful to the reviewer for pointing this out. In our view, this section is needed to illustrate how intrinsic disorder can be distributed within the individual TPR proteins and also to show correlation between predicted disorder and know structural and functional information for highly disordered TPR proteins. We do not think that any kind meta-analysis is needed here, since this section mostly serve an illustration purpose.

  1. It is stated in the abstract that PTMs in TPR proteins are associated with the IDRs. Is there a global analysis that supports this or is this coming from the anecdotal analysis of the 10 most disordered proteins? I see some PTMs in the order regions of the D2P2 analysis.

REPLY:  We are thankful to the reviewer for pointing this out. We did not conduct a specific global analysis of the association of PTMs in TPR proteins with their IDRs. Conclusion mostly comes from the anecdotal analysis of the 10 most disordered proteins. The reviewer is right, and in the D2P2 plots, some PTMs can be located within the ordered regions. However, one should keep in mind that D2P2 plots show only regions, for which predicted disorder score exceed 0.5 threshold, whereas regions with increased flexibility (i.e., those with the disorder scores ranging from 0.2 to 0.5). In fact, the abstract states that IDPRs often serve as targets of various posttranslational modifications, which is not an equivalent of the assumption that all PTMs in TPR proteins are always associated with the IDRs.    

MINOR COMMENTS

  1. I am unclear what “reactions” are affected by TRP repeat proteins. Is this referring to protein binding as a reaction or is there some enzymatic reactions being regulated?

REPLY:  We are thankful to the reviewer for pointing this out. To avoid confusion, we changed “reaction” to “interaction with binding partners”.

  1. I am not clear what the definition here is for “stacking” of TPR repeats since the repeats are offset by a helical twist. All tandem repeats follow each other with some contact. Is this aromatic stacking or do all tandem repeats stack?

REPLY:  We are thankful to the reviewer for pointing this out. We provided the following clarification and the related reference to the revised manuscript: “where the helices within each repeat stack together with helices in adjacent TPRs to form a right-handed superhelix”

  1. I am failing to appreciate the remarkableness of TPR designability and engineering. There are many protein scaffolds that have been engineered for alternate specificity and higher affinity. Why does it matter that these repeats can be concatenated unnaturally without limit?  I would suggest toning down language about exceptionality unless directly justified in the text.

REPLY:  We are thankful to the reviewer for pointing this out. We toned done the corresponding statements.

  1. What is an absolute majority?

REPLY:  We are thankful to the reviewer for pointing this out. We removed the word absolute.

  1. MoRFs are not well introduced before they are analyzed. I am unclear how these are being predicted in the current work.

REPLY:  We are thankful to the reviewer for pointing this out. Description of the MoRF prediction is added to the Materials and Methods. They were identified by the ANCHOR algorithm.

TYPOS

  1. The Figure 1 legend should read “…ability [of] TPR domains to be stack continuously one upon the other.”

REPLY:  This was reworded as described

  1. The colors in Figure 6 are not in agreement with the description in the main text.

REPLY:  We corrected the figure description to correctly describe the color representation in the new plots

  1. The discussion of Figure 13 is jumbled. The text states that Figure 13A identifies the location of TPR repeats in the N-termnus but the numbers quoted are in the middle of the protein.  The sentence appears to be missing a clause as it ends with “and.”

REPLY:  We removed “and.” and reworded the description for clarity

Reviewer 2 Report

In this manuscript, Van Bibber et al. presents a study wherein a systematic analysis of the functionally diverse motif Tetratricopeptide repeats (TPRs) was conducted to characterize the intrinsic disorder properties of TPR-containing proteins. Combining disordered region predictor algorithms (i.e., PONDR VLXT, PONDR VSL2 and PONDR FIT) and the protein-protein interaction identifier program STRING, the researchers observed that TPR-containing proteins vary in their levels of intrinsic disorder and that these structurally flexible regions partake in various protein interactions and post-translational modifications.

The following are this reviewer’s MAJOR comments/suggestions/questions regarding the results/interpretations reported and discussed in the manuscript:

  1. Using 3 different PONDR disorder predictors, the authors analyzed 166 human TPR protein sequences (Fig. 3A, B and C). While the results between the PONDR predictors are glaringly different, most dramatic between the PONDR FIT results vs. the other two (VLXT and VSL2), this apparent discrepancy is not discussed in the main text. Related to this discrepancy is the question, “how accurate are these disorder predictors?”
  2. If the cutoffs used to classify proteins as highly ordered, or moderately or highly disordered are described in the text to be arbitrary, what is preventing a “chicken and egg” scenario? That is, are the proteins really predominantly more disordered, or are the cutoffs too generous in classifying proteins as disordered?

Below are MINOR comments:

  1. Readability of the manuscript can be improved with additional editing to fix grammatical errors and reduce the number of run-on sentences. For example, regarding the introduction, the clarity of the key points was often lost by the long, complex sentence structures. Careful selection of words as well as condensing the long sentences will improve the clarity.
  2. 2 legend x-axis should state that numbers represent # residues
  3. 3 disorder score is in 0-100% scale but discussed in text as fraction 0-1. For consistency, the authors should choose one of these scales.
  4. 6 color coding inconsistent with the legend description. Some of the axes labels have obstructed view due to the color legends.
  5. In reference to Figure 7, the authors numbered the quadrants in a clockwise direction starting with Q1 in the lower right and ending with Q4 in the upper right. Perhaps, applying the conventional method starting with Q1 in the upper right and labeling in a counterclockwise fashion will assist interpretation of the graph.
  6. Regarding Section 2.4 “Structural Properties of Human TPR proteins,” the text mentioned the structural mosaic of human TPR proteins, and while it did briefly highlight the prominence of α-helices, the analysis can be substantiated by findings of other common structural moieties. In addition, Figure 12 can also benefit from this addition if circles or arrows were added to guide readers to the structural similarities.
  7. In the supplementary data, the final table should also contain a figure caption that summarizes the key takeaways and guide the reader.

Author Response

In this manuscript, Van Bibber et al. presents a study wherein a systematic analysis of the functionally diverse motif Tetratricopeptide repeats (TPRs) was conducted to characterize the intrinsic disorder properties of TPR-containing proteins. Combining disordered region predictor algorithms (i.e., PONDR VLXT, PONDR VSL2 and PONDR FIT) and the protein-protein interaction identifier program STRING, the researchers observed that TPR-containing proteins vary in their levels of intrinsic disorder and that these structurally flexible regions partake in various protein interactions and post-translational modifications.

REPLY: We are thankful to this reviewer for careful reading of the manuscript, high evaluation of our work and constructive criticism. We tried to address all the queries and hope that the revised manuscript became more suitable for publication.

The following are this reviewer’s MAJOR comments/suggestions/questions regarding the results/interpretations reported and discussed in the manuscript: 

  1. Using 3 different PONDR disorder predictors, the authors analyzed 166 human TPR protein sequences (Fig. 3A, B and C). While the results between the PONDR predictors are glaringly different, most dramatic between the PONDR FIT results vs. the other two (VLXT and VSL2), this apparent discrepancy is not discussed in the main text. Related to this discrepancy is the question, “how accurate are these disorder predictors?”

REPLY: We are thankful to this reviewer for pointing this out. The use multiple computational tools for prediction of intrinsic disorder in proteins is an accepted practice in the field. Since different computational tools use different attributes and models for finding disordered residues and regions, it is a common situation, when different tools will generate rather different outputs. There is no an accepted consensus of which disorder predictor is the best in evaluating disorder predisposition of proteins. In reality, different tools are sensitive to different disorder-related aspects. In reality, different tools are sensitive to different disorder-related aspects and therefore all of them contain some useful information.  Corresponding clarification is added to the revised manuscript.

  1. If the cutoffs used to classify proteins as highly ordered, or moderately or highly disordered are described in the text to be arbitrary, what is preventing a “chicken and egg” scenario? That is, are the proteins really predominantly more disordered, or are the cutoffs too generous in classifying proteins as disordered?

REPLY: We are thankful to the reviewer for pointing this out. ADS and PPDR were introduced for the quantification of disorder status of large protein sets. Although these numbers could be not too informative for a given protein, they do show a general picture for large datasets. As it is indicated in the text explaining Figure 5, these two measures are not identical, since, theoretically, a protein with the PPDR of 100% might have the ADS ranging from 0.5 to 1.0; whereas a protein with the PPDR of 0% might have any ADS < 0.5. The cutoff ADS and PPDR values for classification of proteins as highly ordered, moderately and highly disordered are arbitrary. For PPDR, the cutoff values were proposed almost a decade ago (Rajagopalan, K.; Mooney, S.M.; Parekh, N.; Getzenberg, R.H.; Kulkarni, P. A majority of the cancer/testis antigens are intrinsically disordered proteins. J Cell Biochem 2011, 112, 3256-3267, doi:10.1002/jcb.23252, which is our reference #73) and are rather intensively used in the literature since then. Therefore, we want to emphasize here that we did not introduced the cutoffs used in this study to classify proteins as highly ordered, or moderately or highly disordered. Instead, we used the values which are commonly utilized in the field.

Below are MINOR comments:

  1. Readability of the manuscript can be improved with additional editing to fix grammatical errors and reduce the number of run-on sentences. For example, regarding the introduction, the clarity of the key points was often lost by the long, complex sentence structures. Careful selection of words as well as condensing the long sentences will improve the clarity.

REPLY:  We are thankful to the reviewer for pointing this out. We edited the entire manuscript for grammar and readability.

  1. 2 legend x-axis should state that numbers represent # residues
  2. 3 disorder score is in 0-100% scale but discussed in text as fraction 0-1. For consistency, the authors should choose one of these scales.

REPLY:  We are thankful to the reviewer for pointing this out.  We reworded part of the text to make clearer we are discussing two disorder scores. Average disorder is on the same 0-1 scale as the per-residue predictions, and percent disorder that is discussed as a percentage. Also, we removed the percentages from the left-side of Figure 3 charts since they were incorporated in the pie chart anyway. They were just representing the proportion of our sample represented by each column, but may have been causing confusion with percent disorder scores.

  1. 6 color coding inconsistent with the legend description. Some of the axes labels have obstructed view due to the color legends.

REPLY:  We are thankful to the reviewer for pointing this out. We corrected the figure description to correctly describe the color representation in the new plots. Legends cannot be moved

  1. In reference to Figure 7, the authors numbered the quadrants in a clockwise direction starting with Q1 in the lower right and ending with Q4 in the upper right. Perhaps, applying the conventional method starting with Q1 in the upper right and labeling in a counterclockwise fashion will assist interpretation of the graph.

REPLY:  We are thankful to the reviewer for pointing this out. We are following here the procedure used in previous publications.

  1. Regarding Section 2.4 “Structural Properties of Human TPR proteins,” the text mentioned the structural mosaic of human TPR proteins, and while it did briefly highlight the prominence of α-helices, the analysis can be substantiated by findings of other common structural moieties. In addition, Figure 12 can also benefit from this addition if circles or arrows were added to guide readers to the structural similarities.

REPLY:  We are thankful to the reviewer for pointing this out. Corresponding information was added to the revised manuscript.

  1. In the supplementary data, the final table should also contain a figure caption that summarizes the key takeaways and guide the reader.

REPLY:  We are thankful to the reviewer for pointing this out. The corresponding information was added.

Round 2

Reviewer 1 Report

I have no scientific or technical concerns with this manuscript.

Reviewer 2 Report

The manuscript may be accepted as is.